# Typhoid toxin sorting and exocytic transport from *Salmonella* Typhi-infected cells

**Shu-Jung Chang[1,2], Yu-Ting Hsu[2], Yun Chen[2], Yen-Yi Lin[2], Maria Lara-Tejero[1], Jorge E Galan[1]\***

[1]Department of Microbial Pathogenesis, Yale University School of Medicine, New Haven, United States; [2]Graduate Institute of Microbiology, College of Medicine, National Taiwan University, Taipei, Taiwan

**Abstract** Typhoid toxin is an essential virulence factor for *Salmonella* Typhi, the cause of typhoid fever in humans. This toxin has an unusual biology in that it is produced by *Salmonella* Typhi only when located within host cells. Once synthesized, the toxin is secreted to the lumen of the *Salmonella*-containing vacuole from where it is transported to the extracellular space by vesicle carrier intermediates. Here, we report the identification of the typhoid toxin sorting receptor and components of the cellular machinery that packages the toxin into vesicle carriers, and exports it to the extracellular space. We found that the cation-independent mannose-6-phosphate receptor serves as typhoid toxin sorting receptor and that the coat protein COPII and the GTPase Sar1 mediate its packaging into vesicle carriers. Formation of the typhoid toxin carriers requires the specific environment of the *Salmonella* Typhi-containing vacuole, which is determined by the activities of specific effectors of its type III protein secretion systems. We also found that Rab11B and its interacting protein Rip11 control the intracellular transport of the typhoid toxin carriers, and the SNARE proteins VAMP7, SNAP23, and Syntaxin 4 their fusion to the plasma membrane. Typhoid toxin's cooption of specific cellular machinery for its transport to the extracellular space illustrates the remarkable adaptation of an exotoxin to exert its function in the context of an intracellular pathogen.

**\*For correspondence:**
jorge.galan@yale.edu

**Competing interest:** The authors declare that no competing interests exist.

## Editor's evaluation

This paper is of interest to microbiologists as well as eukaryotic cell biologists interested in vesicular trafficking pathways. The authors identify several eukaryotic proteins required for typhoid toxin export from *Salmonella* Typhi-infected cells to the extracellular space including mannose-6-phosphate Receptor that serves sorting receptor.

## Introduction

*Salmonella enterica* serovar Typhi is the cause of typhoid fever in humans, a systemic disease that affects approximately 20 million people every year resulting in approximately 150,000 deaths worldwide (*Buckle et al., 2012*; *Dougan and Baker, 2014*; *Kim et al., 2017*; *Mogasale et al., 2014*; *Parry et al., 2002*). A related pathogen, *S. enterica* serovar Paratyphi A, causes a very similar disease in humans (paratyphoid fever) and is increasingly becoming more prevalent in certain areas of the world. Both serovars are restricted to the human host and no animal reservoir for these pathogens has been identified. The emergence of multiple-antibiotic resistant strains of these pathogens, which are becoming endemic in different parts of the world, is a major concern as it is complicating the available

therapies (*Chau et al., 2007*). Although there are several vaccines available that confer partial protection to *S.* Typhi infection, there are no vaccines available for *S.* Paratyphi A (*Milligan et al., 2018*; *Zuckerman et al., 2017*).

Typhoid toxin is a virulence factor that is encoded by both, *S.* Typhi and *S.* Paratyphi A, but that is largely absent from other *S. enterica* serovars that cause self-limiting gastroenteritis (i.e. non-typhoidal serovars) (*Fowler and Galán, 2018*; *Fowler, 2019*; *Galán, 2016*; *Haghjoo and Galán, 2004*; *Song et al., 2013*; *Spanò et al., 2008*). Homologs of the toxin are also present in a very limited number of serovars belonging to clade B that are rarely associated with disease in humans (*Cheng and Wiedmann, 2019*), although these homologs exhibit different biological activity (*Lee et al., 2020*). Typhoid toxin is an AB toxin with a very unusual architecture. Rather than one enzymatic 'A' subunit as seen in all other AB$_5$ toxins, typhoid toxin has two covalently linked A subunits (PltA and CdtB) linked to a pentameric 'B' subunit alternatively composed of PltB or PltC (*Fowler, 2019*; *Song et al., 2013*). PltA is an ADP-ribosyl transferase with an unknown cellular target, and CdtB is an atypical deoxyribonuclease, which inflicts DNA damage on the intoxicated cells. Consistent with the human host specificity of both, *S.* Typhi and *S.* Paratyphi A, the PltB subunit of typhoid toxin specifically recognizes acetyl neuraminic acid (Neu5Ac)-terminated sialoglycans, which are almost unique to the human host since most other mammals display glycolyl neuraminic acid (Neu5Gc)-terminated glycans (*Deng et al., 2014*; *Song et al., 2013*). A human volunteer study suggested that typhoid toxin does not appear to be required for the initiation of *S.* Typhi infection (*Gibani et al., 2019*). However, administration of typhoid toxin into animals engineered to display Neu5Ac-terminated sialoglycans reproduced many of the symptoms of severe typhoid, including various neurological symptoms, stupor, and profound leukopenia (*Deng et al., 2014*; *Fowler, 2019*; *Song et al., 2013*; *Yang et al., 2018*), aspects of the disease that for ethical reasons could not be captured in the human volunteer study (*Gibani et al., 2019*).

A unique aspect of typhoid toxin is that it is only produced when *S.* Typhi is located within mammalian cells (*Fowler and Galán, 2018*; *Spanò et al., 2008*). Once produced, the toxin is secreted from the bacteria to the lumen of the *Salmonella*-containing vacuole (SCV) through a novel dedicated protein secretion system (*Geiger et al., 2020*; *Geiger et al., 2018*; *Hodak and Galán, 2012*), which has been recently proposed to be classified as type X (*Palmer et al., 2021*). The toxin is then packaged within vesicle carriers, which transport it to the extracellular space (*Chang et al., 2016*; *Spanò and Galán, 2012*). The toxin then binds its specific Neu5Ac-bearing receptors, and after receptor-mediated endocytosis, it is transported to its intracellular destination via a specialized retrograde transport pathway (*Chang et al., 2019*). There is no pathway that can transport the toxin from its site of production within the SCV to its place of action within the same cell and therefore intoxication always requires the export of the toxin to the extracellular space (*Spanò et al., 2008*). Very little is known about the exocytic pathway that transports the toxin from the SCV to the extracellular space. A single amino acid substitution in the glycan-binding site of PltB results in a toxin that remains trapped within the SCV after its secretion from the bacteria (*Chang et al., 2016*). This observation suggests that toxin export requires a sorting event most likely mediated by a glycosylated receptor on the luminal side of the *S.* Typhi-containing vacuolar membrane. However, the identity of such putative receptor is unknown.

Here, we report the identification of the typhoid-toxin sorting receptor and components of the cellular machinery involved in toxin-sorting, packaging, and exocytic transport to the extracellular space.

## Results

### Identification of the cation-independent mannose-6-phosphate receptor as the typhoid toxin cargo receptor for sorting into vesicle transport carriers

To identify the putative typhoid toxin sorting receptor, we sought to identify typhoid-toxin interacting proteins in cultured human epithelial cells by affinity purification and liquid chromatography-tandem mass spectrometry (LC-MS/MS) analysis (*Figure 1a*). As a negative control we used a mutant version of typhoid toxin that carries a single amino acid substitution in its PltB subunit (PltB$^{S35A}$). This mutant cannot be packaged into vesicle carrier intermediates because this residue is critical for its interaction

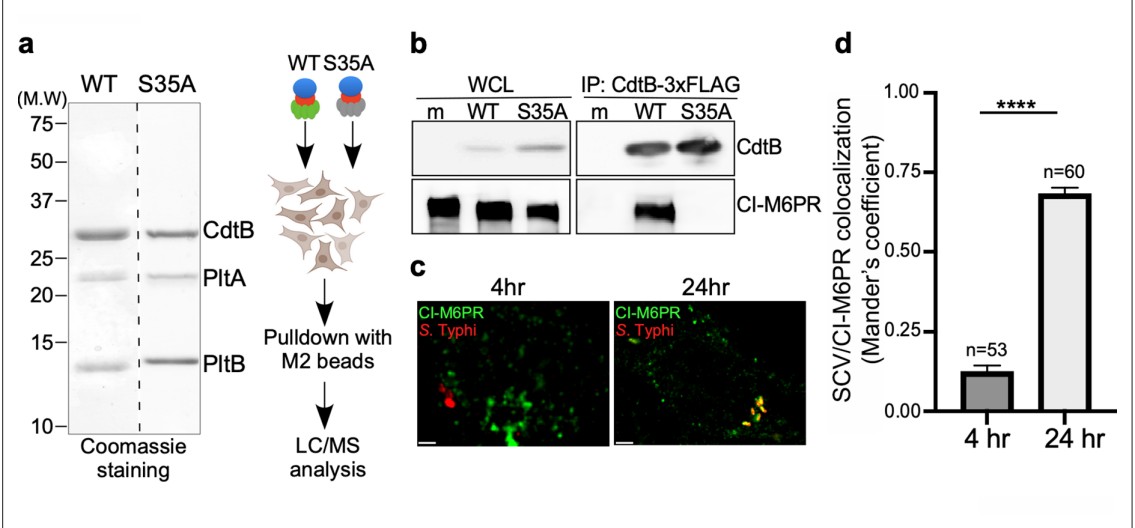

**Figure 1.** Identification of CI-M6PR as a typhoid toxin-interacting protein. (**a and b**) Purified FLAG-tagged wild-type typhoid toxin or its PltB[S35A] mutant unable to bind glycosylated receptor proteins (**a**) were used in affinity purification experiments (outlined in a) to identify typhoid toxin-interacting proteins, which led to the identification of CI-M6PR (see **Figure 1—source data 2**). (**b**) The interaction between typhoid toxin and CI-M6PR was verified in *Salmonella* Typhi-infected cells. Henle-407 cells were infected with *S.* Typhi expressing FLAG-tagged CdtB for 24 hr and the interaction between typhoid toxin and endogenous CI-M6PR was probed by affinity purification with a FLAG antibody (directed to the CdtB subunit of typhoid toxin) and western blot (with antibodies to both FLAG and anti-CI-M6PR). (**c and d**) Co-localization of the *S.* Typhi-containing vacuole and CI-M6PR. Henle-407 cells were infected with *S.* Typhi for the indicated times and examined by immunofluorescence with differentially labeled antibodies to *S.* Typhi and CI-M6PR. The quantification of the co-localization is shown in (**d**). Values (Mander's overlap coefficient) represent the degree of co-localization between CI-M6PR and *S.* Typhi and are the mean ± SEM. ****: $p<0.0001$. Scale bar = 5 μm. SCV: *Salmonella*-containing vacuole; CI-M6PR: cation-independent mannose-6-phosphate receptor. WCL: whole-cell lysates; IP: immunoprecipitation.

The online version of this article includes the following source data and figure supplement(s) for figure 1:

**Source data 1.** Unprocessed coomassie stain of the protein gel.

**Source data 2.** Interacting proteins of cation-independent mannose-6-phosphate receptor (CI-M6PR) identified by immunoprecipitation-mass spectrometry (IP-MS).

**Source data 3.** Unprocessed CdtB and cation-independent mannose-6-phosphate receptor (CI-M6PR) western blots.

**Source data 4.** Raw data of *Figure 1d*.

**Figure supplement 1.** Cells infected with either *Salmonella* Typhimurium (top panels) or *S.* Typhi (lower panels) were fixed at 24 hr post infection, stained with antibodies directed to cathepsin D (CatD) (red) and *Salmonella* LPS (green), and examined under a fluorescence microscope for co-localization.

with its putative Neu5Ac-bearing sorting receptor (*Chang et al., 2016*). The cation-independent mannose-6-phosphate receptor (CI-M6PR) (*Gary-Bobo et al., 2007*; *Stalder and Gershlick, 2020*) was identified as a prominent interacting protein with wild-type typhoid toxin but not with its PltB[S35A] mutant control (*Figure 1—source data 2*). To validate this interaction, we infected cultured human epithelial cells with *S.* Typhi strains expressing FLAG-epitope-tagged versions of wild-type typhoid toxin or its PltB[S35A] mutant form and examined their interaction with CI-M6PR by immunoprecipitation and western blot analysis. Similar to the in vitro experiments, we detected the interaction of CI-M6PR with wild-type typhoid toxin but not with the PltB[S35A] mutant version (*Figure 1b*). Consistent with these observations, we found that CI-M6PR is recruited to the *S.* Typhi-containing vacuole, particularly later in infection (*Figure 1c and d* and *Figure 1—source data 4*). Recruitment of CI-M6PR to the *S.* Typhi-containing vacuole was not due to its delivery to lysosomes since we did not detect the presence of the lysosomal hydrolases on the *S.* Typhi-containing vacuole (*Figure 1—figure supplement 1*). These observations are intriguing since it is well established that CI-M6PR is not recruited to the vacuoles that contain *S.* Typhimurium (*Garcia-del Portillo and Finlay, 1995*; *McGourty et al., 2012*), which does not encode typhoid toxin. Taken together, these results show that CI-M6PR is recruited to the *S.* Typhi-containing vacuoles in cultured human epithelial cells where it interacts with typhoid toxin through its PltB subunit.

To specifically examine the contribution of CI-M6PR to typhoid toxin sorting into vesicle carriers, we generated a CI-M6PR-deficient cell line by CRISPR/Cas9 genome editing (*Figure 2a*). We found that inactivation of CI-M6PR did not affect the ability of *S.* Typhi to gain access to and replicate within these cells (*Figure 2b* and *Figure 2—source data 2*), neither it affected the ability of typhoid toxin to be internalized and intoxicate these cells when exogenously applied (*Figure 2c* and *Figure 2—source data 2*). These results indicate that inactivation of CI-M6PR did not affect cellular processes that may grossly alter *S.* Typhi and/or toxin biology in a manner that may affect the interpretation of these experiments. We then assessed the formation of typhoid toxin vesicle carrier intermediates after *S.* Typhi infection of CI-M6PR-deficient cells. We found in CI-M6PR-deficient cells relative to the parental cell line a marked reduction in the number of typhoid toxin vesicle carrier intermediates that can be visualized as typhoid toxin-associated fluorescent puncta (*Figure 2d and e*, *Figure 2—figure supplement 1*, *Figure 2—figure supplement 2*, and *Figure 2—source data 2*). Consistent with a decreased formation of typhoid toxin carrier intermediates, we also found a reduced level of typhoid toxin in the extracellular medium of *S.* Typhi-infected CI-M6PR-deficient cells (*Figure 2f–h*, *Figure 2—figure supplement 2*, and *Figure 2—source data 2*). It has been previously reported that interference with CI-M6PR recycling (*Ikeda et al., 2008*) or infection with *Salmonella* Typhimurium (*McGourty et al., 2012*; *Selkrig et al., 2020*) results in the secretion of lysosomal enzymes to the extracellular space. However, we found no evidence for the presence of extracellular protease activity that could account for the reduced levels of typhoid toxin in the extracellular media of *S.* Typhi-infected CI-M6PR -/- cells (*Figure 2—figure supplement 3*). Taken together, these results indicate that CI-M6PR serves as a packaging receptor for typhoid toxin sorting from the lumen of SCVs into vesicle export carriers.

## Typhoid toxin export requires the specific intracellular vacuolar compartment that harbors *S.* Typhi

It is well established that, as an intracellular pathogen, *Salmonella* builds its own vacuolar compartment shaped by the activity of its arsenal of effector proteins delivered by its two type III secretion systems (*Galán, 2001*; *Jennings et al., 2017*). Recent work has also shown that there are important differences in the composition of the intracellular vacuole harboring typhoidal (e.g. *S.* Typhi) and non-typhoidal (e.g. *S.* Typhimurium) *S. enterica* serovars. For example, while the Rab-family GTPases Rab29, Rab32, and Rab38 are robustly recruited to the *S.* Typhi-containing vacuole, these GTPases are absent in the vacuolar compartment harboring *S.* Typhimurium (*Spanò and Galán, 2012*; *Spanò et al., 2011*). This is due to differences in the repertoire of type III-secreted effector proteins encoded by these two bacteria. Specifically, *S.* Typhimurium prevents the recruitment of Rab29, Rab32, and Rab38 by delivering of two effectors, GtgE and SopD2, which are absent from *S.* Typhi and target these Rab GTPases with specific protease and GAP activities, respectively (*Spanò et al., 2016*; *Spanò et al., 2011*). It is also well established that the *S.* Typhimurium-containing vacuole does not recruit CI-M6PR, and that the lack of recruitment of this lysosomal marker is also linked to the activity of type III-secreted effectors (*McGourty et al., 2012*; *Figure 3a*, *Figure 3—figure supplement 1*, and *Figure 3—source data 1*). In contrast, we found that the CI-M6PR is recruited to the *S.* Typhi-containing vacuole where it serves as a sorting receptor for typhoid toxin export (*Figures 3a and 1d* and *Figure 3—figure supplement 1*). These findings suggest that specific features of the *S.* Typhi-containing vacuole determined by the type-III-secreted effector repertoire are essential for typhoid toxin export.

To further explore this hypothesis, we cloned the entire *S.* Typhi genomic islet that encodes typhoid toxin into *S.* Typhimurium and examined typhoid toxin export in cells infected with the resulting strain. We reasoned that differences in the vacuolar environments of *S.* Typhimurium may prevent the export of typhoid toxin when expressed in this strain. Cells infected with the *S.* Typhimurium strain expressing typhoid toxin showed markedly reduced levels of vesicle carrier intermediates when compared to *S.* Typhi-infected cells, despite equivalent levels of toxin expression (*Figure 3b*, *Figure 3—figure supplement 2*, and *Figure 3—source data 1*). This is consistent with the observation that the typhoid toxin receptor CI-M6PR is not recruited to the *S.* Typhimurium-containing vacuole (*Figure 3a*, *Figure 3—figure supplement 1*, *Figure 3—source data 1*). These observations are also consistent with the notion that typhoid toxin packaging into vesicle carrier intermediates requires the specific environment of the *S.* Typhi-containing vacuole.

Since the nature of the SCV is strictly dependent on the function of its type III secretion systems (*Galán, 2001*; *Jennings et al., 2017*), we reasoned that differences in the composite of effector

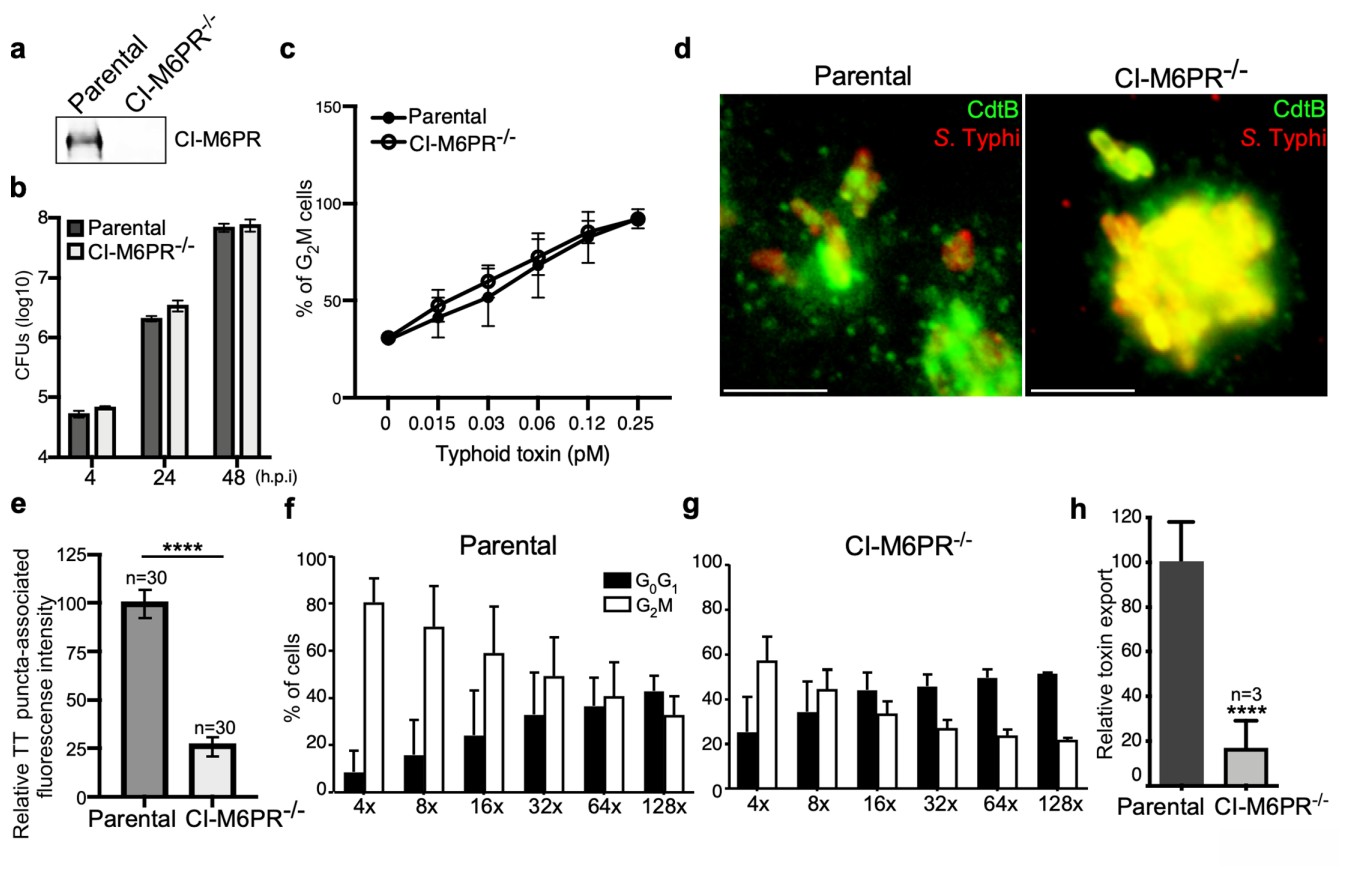

**Figure 2.** CI-M6PR is the typhoid toxin cargo receptor for packaging into vesicle transport carriers. (**a**) Immunoblot analyses of CI-M6PR expression in parental HEK293T and CI-M6PR-deficient cells generated by CRISPR/Cas9 genome editing. (**b**) Intracellular survival of *Salmonella* Typhi in parental HEK293T and CI-M6PR-deficient cells. Cells were infected with *S.* Typhi with a multiplicity of infection of 30, and CFUs were enumerated at 4, 24, and 48 hr after infection. Values are the mean ± SD of three independent experiments. CFU = colony-forming units; SD = standard deviation. (**c**) Toxicity of typhoid toxin in parental HEK293T and CI-M6PR-deficient cells. Cells were treated with a serial dilution of purified typhoid toxin and the typhoid toxin intoxication was evaluated by examining the proportion of cells in G2/M as a consequence of typhoid toxin-mediated DNA damage. The data shown are the mean ± SD of three independent experiments. (**d and e**) Typhoid toxin transport carrier formation in parental HEK293T and CI-M6PR-deficient cells. Cells were infected with a *S.* Typhi strain expressing 3xFLAG epitope-tagged CdtB and stained with antibodies against the FLAG epitope (green) and *S.* Typhi LPS (red) (**d**). Scale bar, 5 µm. The quantification of typhoid toxin-associated fluorescent puncta, a measure of typhoid toxin carrier intermediates in infected cells, is shown in (**e**). Values represent relative fluorescence intensity and are the mean ± SEM of one of three independent experiments. ****: p<0.0001, unpaired two-sided *t* test. The results of two additional experiments are shown in *Figure 2—figure supplement 1*. (**f–h**) Quantification of typhoid toxin export into the infection medium. Infection media obtained from *S.* Typhi-infected HEK293T parental (**f**) or CI-M6PR-deficient cells (**g**) were serially diluted as indicated and applied to uninfected HEK293T cells. The cell cycle profile of treated cells was analyzed by flow cytometry, and the percentage of cells at the G2/M phase, a measure of typhoid toxin toxicity, was determined. Values are the mean ± SD of three independent experiments. The relative toxicity of the different samples, shown in (**h**), was measured by determining the percentage of cells in the G2/M phase from the results of the dilution of infection media experiments (shown in **f** and **g**) fitted by nonlinear regression. Values were normalized relative to those of the parental cells, which was considered to be 100 and are the mean ± SD of three independent experiments. ****: p<0.0001, unpaired two-sided *t* test. CI-M6PR: cation independent mannose-6-phosphate receptor; TT: typhoid toxin.

The online version of this article includes the following source data and figure supplement(s) for figure 2:

**Source data 1.** Unprocessed CdtB western blot.

**Source data 2.** Raw data of *Figure 2b–h*.

**Figure supplement 1.** Typhoid toxin transport carrier formation in parental HEK293T and cation-independent mannose-6-phosphate receptor (CI-M6PR)-deficient cells.

**Figure supplement 2.** Analysis of independently isolated clones of the CRISP-Cas9-generated CI-M6PR-/- cells shown in *Figure 2*.

**Figure supplement 3.** Analysis for the potential presence of proteases on the *Salmonella* Typhi infection media that could degrade typhoid toxin.

**Figure supplement 3—source data 1.** Unprocessed CdtB, Flag, and actin western blots.

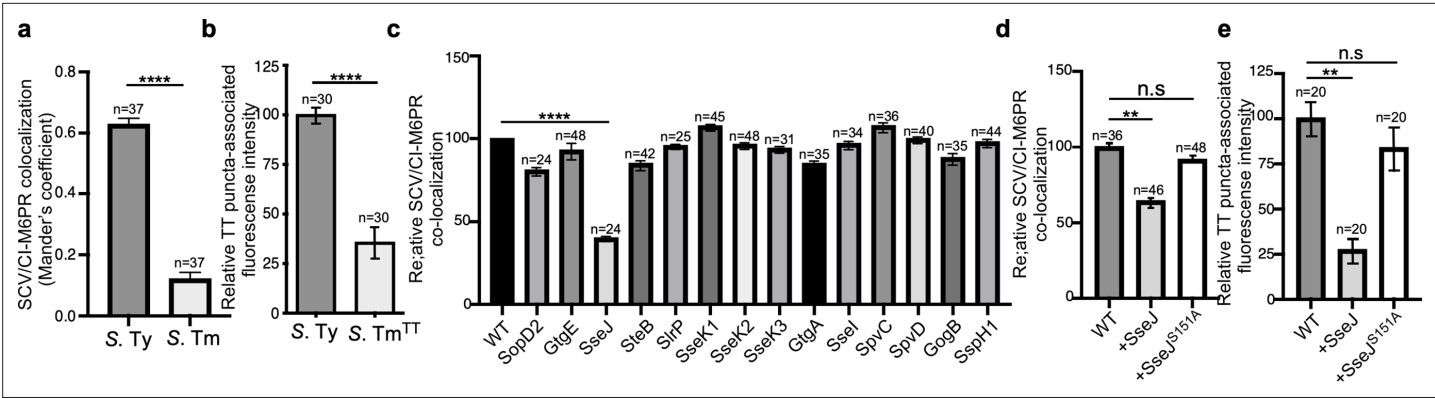

**Figure 3.** Typhoid toxin export requires the specific vacuolar compartment that harbors *Salmonella* Typhi. (**a**) Quantification of CI-M6PR recruitment to the *Salmonella*-containing vacuoles. Cells infected with either *S.* Typhi (*S.* Ty) or *S.* Typhimurium (*S.* Tm) were fixed at 24 hr post infection, stained with antibodies directed to CI-M6PR and *Salmonella* LPS, and examined under a fluorescence microscope for co-localization. Values (Mander's overlap coefficient) represent the degree of co-localization between CI-M6PR and *Salmonella*-containing vacuoles and are the mean ± SEM of one representative experiment of three independent experiments. ****: p<0.0001, unpaired two-sided *t* test. The results of two additional experiments are shown in *Figure 3—figure supplement 1*. (**b**) Quantification of the intensity of typhoid toxin-associated fluorescent puncta associated with typhoid toxin carrier intermediates in infected cells. Cells infected with either *S.* Typhi or the *S.* Typhimurium strain expressing FLAG-tagged CdtB and 24 hr after infection, cells were fixed and stained with differentially labeled antibodies against CdtB and *Salmonella* LPS. Values represent relative fluorescence intensity associated with typhoid toxin carriers and are the mean ± SEM of one representative experiment. ****: p<0.0001, unpaired two-sided *t* test. The results of an additional independent experiment are shown in *Figure 3—figure supplement 2*. (**c**) Quantification of CI-M6PR recruitment to the *Salmonella*-containing vacuoles. Cells infected with *S.* Typhi strains expressing the indicated *S.* Typhimurium effector proteins were fixed at 24 hr post infection and then stained as described above. Values (Mander's overlap coefficient) represent the relative recruitment of CI-M6PR to the SCV and are the mean ± SEM. ****: p<0.0001, two-sided Student's *t* test. The results of an additional independent experiment are shown in *Figure 3—figure supplement 3*. (**d**) Quantification of relative CI-M6PR recruited to *Salmonella*-containing vacuoles in cells infected with *S.* Typhi expressing either wild-type SseJ or its catalytic mutant SseJ[S151A]. Infected cells were fixed and then stained as indicated in (**c**). Values (Mander's overlap coefficient) represent the relative recruitment of CI-M6PR to the SCV and are the mean ± SD. **: p<0.01, unpaired two-sided *t* test. n.s.: difference not statistically significant. The results of two additional independent experiments are shown in *Figure 3—figure supplement 3b*. (**e**) Quantification of the intensity of typhoid toxin-associated fluorescent puncta associated with typhoid toxin carrier intermediates in cells infected with *S.* Typhi expressing either wild-type SseJ or its catalytic mutant SseJ[S151A]. Infected cells were fixed and then stained as described in (**b**). Values represent relative fluorescence intensity and are the mean ± SEM. **: p<0.01, unpaired two-sided *t* test. n.s.: difference not statistically significant. The results of an additional independent experiment are shown in *Figure 3—figure supplement 3c*. SCV: *Salmonella*-containing vacuole; CI-M6PR: cation-independent mannose-6-phosphate receptor; TT: typhoid toxin.

The online version of this article includes the following source data and figure supplement(s) for figure 3:

**Source data 1.** Raw data of *Figure 3a–e*.

**Figure supplement 1.** Quantification of cation-independent mannose-6-phosphate receptor (CI-M6PR) recruitment to the *Salmonella* Typhi *or S.* Typhimurium-containing vacuoles.

**Figure supplement 2.** Typhoid toxin vesicle carrier intermediates in cells infected with *S.* Typhimurium expressing typhoid toxin.

**Figure supplement 3.** Effect of the expression of *Salmonella* Typhimurium effectors on *S.* Typhi's ability to recruit cation-independent mannose-6-phosphate receptor (CI-M6PR) to its vacuolar compartment.

**Figure supplement 4.** Effect of the expression of *Salmonella* Typhimurium effectors on the ability of *S.* Typhi to recruit cation-independent mannose-6-phosphate receptor (CI-M6PR) to its vacuolar compartment.

**Figure supplement 5.** Effect of the expression of *Salmonella* Typhimurium SseJ or its catalytic SseJ[S151A] mutant on *S.* Typhi's ability to recruit cation-independent mannose-6-phosphate receptor (CI-M6PR) to its vacuolar compartment.

**Figure supplement 6.** Levels of CI-M6PR in cells infected with wild-type (WT) S*almonella* Typhi or the same strain expressing the *S.* Typhimurium effector SseJ.

**Figure supplement 6—source data 1.** Unprocessed cation-independent mannose-6-phosphate receptor (CI-M6PR) and tubulin western blots.

**Figure supplement 7.** Effect of the expression of *Salmonella* Typhimurium SseJ or its catalytic SseJ[S151A] mutant on the formation of typhoid toxin transport carrier intermediates.

proteins expressed by *S.* Typhi and *S.* Typhimurium may be ultimately responsible for the differences in the ability of typhoid toxin to be packaged into vesicle carrier intermediates. More specifically, we reasoned that the action of some type III effector(s) encoded by *S.* Typhimurium but absent from *S.* Typhi may preclude the recruitment of CI-M6PR to the SCV and thus prevent the packaging of typhoid

toxin into transport carriers. To test this hypothesis, we individually expressed in *S.* Typhi the *S.* Typhimuirum SPI-1 or SPI-2 T3SS effector proteins that are absent or pseudogenes in *S.* Typhi (*Parkhill et al., 2001*) (i.e. SopD2, GtgE, SseJ, SteB, SlrP, SseK1, SseK2, SseK3, GtgA, SseI, SpvC, SpvD, GogB, and SspH1). We infected cells with the resulting strains and examined the recruitment of CI-M6PR to the *S.* Typhi-containing vacuoles. Of all the effectors tested, only the expression of SseJ resulted in a significant reduction in the recruitment of CI-M6PR to the *S.* Typhi-containing vacuole (*Figure 3c and d*, *Figure 3—figure supplement 3*, *Figure 3—figure supplement 4*, *Figure 3—figure supplement 5* and *Figure 3—source data 1*), without altering the total levels of CI-M6PR in the infected cells (*Figure 3—figure supplement 6*). Consistent with the reduced recruitment of its sorting receptor, the formation of the typhoid toxin carriers in cells infected with the SseJ-expressing *S.* Typhi strain was significantly reduced (*Figure 3e*, *Figure 3—figure supplement 7* and *Figure 3—source data 1*). SseJ has been shown to modify the lipid composition of the SCV by esterifying cholesterol through its glycerophospholipid:cholesterol acyltransferase activity (*Kolodziejek and Miller, 2015*; *Ohlson et al., 2005*). In addition, in a catalytic-independent manner, SseJ recruits the eukaryotic lipid transporter oxysterol-binding protein 1 to the SCV, thus contributing to the integrity of this compartment (*Kolodziejek et al., 2019*). To investigate which of these activities is responsible for the prevention of the recruitment of CI-M6PR to the *S.* Typhi-containing vacuole, we expressed a catalytic mutant of SseJ (SseJ$^{S151A}$) in *S.* Typhi and examined the recruitment of CI-M6PR to the *S.* Typhi-containing vacuole. We found that in contrast to wild type, expression of SseJ$^{S151A}$ had no effect in the recruitment of CI-M6PR to the *S.* Typhi-containing vacuole, and on the formation of typhoid toxin vesicle carrier intermediates (*Figure 3e*, *Figure 3—figure supplement 5* and *Figure 3—figure supplement 7*, and *Figure 3—source data 1*), indicating that the modification of the lipid composition of the SCV through SseJ's catalytic activity influences CI-M6PR recruitment. Taken together, these results indicate that differences in the effector protein composition dictates the ability of *S.* Typhi to intersect with the CI-M6PR sorting receptor and the subsequent packaging of typhoid toxin into vesicle carriers intermediates essential for its export to the extracellular space.

## COPII mediates the formation of typhoid toxin export carriers

To investigate potential mechanisms by which the typhoid toxin export carriers are formed, we sought to identify specific coat proteins and cargo adaptors that promote the budding of toxin carriers from the SCV. Vesicle coats that initiate the budding process and direct each vesicle to its destination are most often comprised of multiple subunits (*Dell'Angelica and Bonifacino, 2019*; *McMahon and Mills, 2004*). We specifically examined the potential contribution to typhoid toxin packaging of four well-characterized coat or adaptor proteins: clathrin (*Briant et al., 2020*), coat protein complex II (COPII) (*McCaughey and Stephens, 2018*), and adaptor-related protein complex 3 (AP3) and 4 (AP4) (*Hirst et al., 2013*; *Odorizzi et al., 1998*). We focused on these coat proteins because they carry out their function at distinct compartments within the secretory pathway. Using CRISPR/Cas9 genome editing, we generated HEK293T cells defective for CLTC (clathrin heavy chain), SEC23B (an inner coat protein of COPII), AP3B1 (AP3 subunit beta-1), and AP4M1 (AP4 subunit mu-1) (*Figure 4—figure supplement 1a*), whose deficiency abrogates the assembly of their respective vesicle coats. We found that typhoid toxin expression as well as the toxicity of exogenously administered typhoid toxin was unaltered in all these cell lines (*Figure 4a* and *Figure 4—figure supplement 1b*) indicating that the absence of the different coats did not affect the endosomal system in a manner that could interfere with the interpretation of the results. We infected the parental and mutant cell lines with *S.* Typhi-expressing epitope-tagged CdtB and examined the formation of vesicle carrier intermediates by fluorescence microscopy, and the export of the toxin to the extracellular medium. Cells defective in CLTC, AP3B1, or AP4M1 showed no defect in the formation of toxin carrier intermediates although AP4M1-deficient cells showed decreased toxin export (*Figure 4b–d*, *Figure 4—figure supplement 2*, *Figure 4—figure supplement 3*, *Figure 4—figure supplement 4*, and *Figure 4—source data 2*). In contrast, SEC23B-deficient cells showed a significant decrease in the formation of typhoid toxin carriers and reduced toxin export (*Figure 4b–d*, *Figure 4—figure supplement 1*, *Figure 4—figure supplement 4* and *Figure 4—source data 2*). Recruitment of the packaging receptor CI-M6PR to the *S.* Typhi-containing vacuoles in SEC23B-deficient cells was indistinguishable from the recruitment observed in the parental cells (*Figure 4—figure supplement 5* and *Figure 4—figure supplement 5—source data 1*), indicating that the defect in typhoid toxin packaging was not due to the impaired recruitment of CI-M6PR.

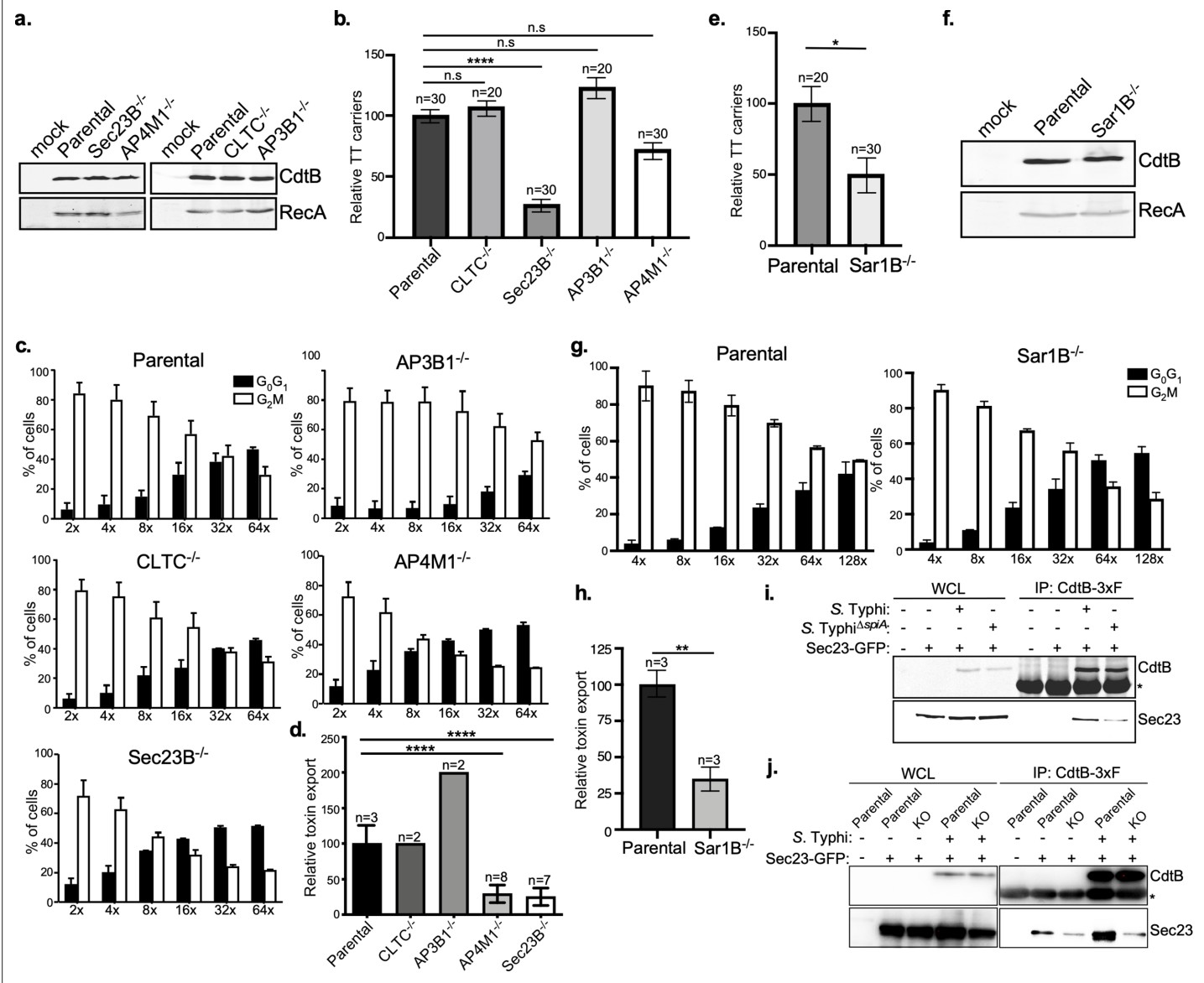

**Figure 4.** COPII drives the formation of typhoid toxin export carriers. (**a**) Western blot analysis of the expression of typhoid toxin in parental HEK293T cells and derivatives deficient in SEC23B (an inner coat protein of COPII), AP4M1 (AP4 subunit Mu-1), CLTC (clathrin heavy chain), or AP3B1 (AP3 subunit Beta-1). Cells were infected with S. Typhi expressing FLAG-tagged CdtB and 24 hr after infection, cells were lysed and analyzed by western blot with antibodies directed to the FLAG epitope and the *Salmonella* protein RecA. (**b**) Quantification of the intensity of fluorescent puncta associated with typhoid toxin carrier intermediates in parental HEK293T and the indicated deficient cells. Values represent relative fluorescence intensity and are the mean ± SEM. ****: p<0.0001, unpaired two-sided *t* test. n.s.: difference not statistically significant. The results of an additional independent experiment are shown in ***Figure 4—figure supplement 2***. (**c and d**) Quantification of typhoid toxin export into the infection medium. Infection media obtained from *S*. Typhi-infected HEK293T parental and the indicated deficient cells were serially diluted as indicated and applied to uninfected HEK293T cells. The cell cycle profile of treated cells was analyzed by flow cytometry, and the percentage of cells in the G2/M phase, a measure of typhoid toxin toxicity, was determined. Values are the mean ± SD of three independent experiments. The relative toxicity of the different samples, shown in (**d**), was measured by determining the percentage of cells in the G2/M phase from the results of the dilution of infection media experiments (shown in c) fitted by nonlinear regression. Values were normalized relative to those of the parental cells, which was considered to be 100 and are the mean ± SD of three independent experiments. ****: p<0.0001, unpaired two-sided *t* test. (**e**) Quantification of the intensity of fluorescent puncta associated with typhoid toxin carrier intermediates in parental HEK293T and Sar1B-deficient cells. Values represent relative fluorescence intensity and are the mean ± SEM. *: p<0.05, unpaired two-sided *t* test. The results of an additional independent experiment are shown in ***Figure 4—figure supplement 2***. (**f**) Western blot analysis of the expression of typhoid toxin in parental HEK293T and Sar1B-deficient cells. Cells were infected with *S*. Typhi expressing FLAG-tagged CdtB and the levels of typhoid toxin were evaluated by western blot 24 hr after infection as indicated in (**a**). (**g and h**) Quantification of typhoid toxin export into the infection medium of parental HEK293T and Sar1B-deficient cells after infection with *S*. Typhi. Values are the mean ± SD of three

*Figure 4 continued on next page*

*Figure 4 continued*

independent experiments (**g**). The relative toxin export is shown in (**h**) and was determined as described in (**c**) and (**d**). Values are the mean ± SD of three independent experiments. \*\*: p<0.01, unpaired two-sided *t* test. (**i**) Western blot analysis of the interaction of typhoid toxin with Sec23A. HEK293T cells transiently transfected with a plasmid-expressing GFP-tagged Sec23A were infected with either wild-type *S*. Typhi (multiplicity of infection [MOI] = 30) or its isogenic *spiA* mutant (MOI = 90), which is defective in its SPI-2-encoded type III secretion system. Twenty-four hours after infection, cells were lysed and the interaction between typhoid toxin and Sec23A was probed by immunoprecipitation with anti-FLAG M2 affinity gel and immunoblotting with anti-CdtB and anti-GFP antibodies. (**j**) Western blot analysis of the interaction between typhoid toxin and Sec23A in parental HEK293T and CI-M6PR-deficient cells. Parental HEK293T and CI-M6PR-deficient cells were transiently transfected with a plasmid-expressing GFP-tagged Sec23A followed by infection with wild-type *S*. Typhi as described in (**i**). Cells were lysed at 24 hr post infection and subjected to immunoprecipitation with an anti-FLAG M2 antibody, and immunoblotting with anti-CdtB and anti-GFP antibodies. CI-M6PR: cation-independent mannose-6-phosphate receptor; WCL: whole-cell lysates; IP: immunoprecipitates; KO: knockout; TT: typhoid toxin.

The online version of this article includes the following source data and figure supplement(s) for figure 4:

**Source data 1.** Unprocessed CdtB and RecA western blots.

**Source data 2.** Raw data of *Figure 4b–e and g–h*.

**Source data 3.** Unprocessed CdtB and Sec23 western blots.

**Source data 4.** Unprocessed CdtB and Sec23 western blots.

**Figure supplement 1.** Genotypic and phenotypic characterization of CRISPR/Cas9-generated deficient cell lines.

**Figure supplement 1—source data 1.** Unprocessed western blot and DNA gels of genotyping of indicated cell lines.

**Figure supplement 1—source data 2.** Unprocessed DNA gels of genotyping of indicated cell lines.

**Figure supplement 2.** Independent experiments of *Figure 4b and e*.

**Figure supplement 3.** Analysis of independent clones of the CRISPR-Cas9-generated CLTB- and CLTC-deficient cells shown in *Figure 4*.

**Figure supplement 4.** Analysis of independent clones of the CRISP-Cas9-generated Sec23A-, Sec23B-, AP3B1-, AP4M1-, and Sar1B-deficient cells shown in *Figure 4*.

**Figure supplement 5.** Co-localization of the *Salmonella* Typhi-containing vacuole and cation-independent mannose-6-phosphate receptor (CI-M6PR) in Sec23B-deficient cells.

**Figure supplement 5—source data 1.** Raw data of *Figure 4—figure supplement 5b*.

**Figure supplement 6.** Co-localization of the *Salmonella* Typhi-containing vacuole and Sec23A.

Assembly of the COPII coat requires the small GTPase Sar1, which coordinates the recruitment of the coat subunits to the target membrane thus initiating the budding process (*Cevher-Keskin, 2013*). We therefore investigated the contribution of Sar1 to typhoid toxin packaging into vesicle carriers by examining their presence in Sar1-deficient cells generated by CRISPR/Cas9 genome editing (*Figure 4—figure supplement 1c and d*). We found a marked reduction in the levels of fluorescent puncta associated with toxin carriers in SAR1B-deficient cells infected with *S*. Typhi despite similar expression levels of typhoid toxin (*Figure 4e and f*, *Figure 4—figure supplement 2* and *Figure 4—source data 2*). Consistent with this observation, in comparison to the parental cell line, we found a significant decrease in the levels of typhoid toxin in the infection medium of SAR1B-deficient cells (*Figure 4g and h*, *Figure 4—figure supplement 4*, and *Figure 4—source data 2*). If COPII plays a pivotal role in the formation of typhoid toxin export carriers, we reasoned that we should be able to observe its recruitment to the SCV and the formation of a complex of typhoid toxin and components of the COPII coat. We observed robust recruitment of Sec23 to the SCV, particularly at later times after infection (*Figure 4—figure supplement 6*). To investigate the formation of a complex between typhoid toxin and COPII, HEK293T parental cells as well as CI-M6PR -/- derivatives transiently expressing GFP-tagged Sec23 were infected with wild-type *S*. Typhi or the *ΔspiA* isogenic mutant (as a negative control) expressing FLAG epitope-tagged CdtB. The formation of a typhoid toxin/Sec23 complex was then investigated by co-immunoprecipitation. We found that typhoid toxin could be detected in complex with SEC23 in HEK293T parental cells infected with wild-type *S*. Typhi (*Figure 4i*), although formation of the complex was markedly reduced in cells infected with the *S*. Typhi *ΔspiA* mutant strain (*Figure 4i*). This mutant strain is defective in the SPI-2 T3SS and consequently has an altered intracellular vacuole and is therefore unable to efficiently package typhoid toxin into vesicle transport carriers (*Chang et al., 2016*). Importantly, the typhoid toxin-Sec23 complex could not be detected in cells lacking CI-M6PR, indicating that formation of the complex requires

the presence of the toxin's sorting receptor (*Figure 4j*). Taken together, our results indicate that the packaging of typhoid toxin into vesicle carrier intermediates is dependent on COPII.

## The transport of typhoid toxin vesicle carriers is dependent on Rab11B and its interacting protein RIP11

The transport of cargo proteins destined for targeting to other intracellular compartments or the extracellular space most often entails multiple steps, which requires a specific set of proteins that ensure the accuracy of cargo delivery. Among these proteins are Rab GTPases, which regulate vesicular transport to most cellular compartments, and are therefore excellent candidates to participate in the regulation of the exocytic transport of typhoid toxin (*Pfeffer, 2017*; *Stenmark, 2009*; *Zhen and Stenmark, 2015*). To identify Rab GTPases that could be potentially involved in typhoid toxin transport, we examined typhoid toxin transport in cell lines deficient in a subset of Rab-family GTPases (or its regulators) that have been previously implicated in various exocytic pathways. More specifically, using CRISPR/Cas9 genome editing, we generated cell lines deficient in Rab27A, Rab27B, Rab11A, Rab11B, and HPS4, a component of the BLOC3 complex which is an exchange factor essential for the function of both, Rab32 and Rab38 (*Gerondopoulos et al., 2012*; *Figure 5—figure supplement 1*). The different cell lines were then infected with *S.* Typhi and the levels of typhoid toxin in the infection media, a direct measure of its export, were determined. We found that inactivation of Rab27A, Rab27B, Rab11A, and HPS4 did not alter the transport of typhoid toxin as demonstrated by the presence of equivalent levels of the toxin in the culture media as those observed in the parental cells (*Figure 5a*, *Figure 5—figure supplement 2*, and *Figure 5—source data 1*). In contrast, cells defective in Rab11B (but not in its close homolog Rab11A) exhibited significantly reduced export of typhoid toxin to the extracellular media (*Figure 5a*, *Figure 5—figure supplement 2*, and *Figure 5—source data 1*), even though the levels of toxin expression in the infected cells were indistinguishable from those of the parent cell (*Figure 5b*). Although the absence of Rab11B translated into a slight (not statistically significant) reduction in the immunofluorescence associated with typhoid toxin transport carriers (*Figure 5c*, *Figure 5—figure supplement 2*, *Figure 5—figure supplement 3*, and *Figure 5—source data 1*), this difference cannot account for the observed differences in toxin export (*Figure 5a* and *Figure 5—source data 1*). These results therefore implicate Rab11B in typhoid toxin transport, most likely in events downstream from its packaging into vesicle carrier intermediates.

Most often Rab GTPases modulate vesicular transport by engaging specific downstream effector proteins, which carry out specific functions. In the case of Rab11B in particular, it has been shown that it regulates some exocytic events by interacting with Rab11-family-interacting protein 5 (Rab11-FIP5) also known as Rip11 (*Horgan and McCaffrey, 2009*). This effector has been shown to be involved in regulating vesicular transport from recycling endosomes to the apical plasma membrane, and in particular, in the regulation of insulin granule exocytosis (*Schonteich et al., 2008*; *Sugawara et al., 2009*). It exerts its function at least in part by linking vesicle carriers to kinesin motors (*Schonteich et al., 2008*), which presumably move the vesicles through microtubule tracks. To determine whether Rip11 also participates in toxin export, we generated Rip11-deficent cells by CRISPR/Cas9 gene editing (*Figure 5—figure supplement 1*) and examined typhoid toxin export after *S.* Typhi infection. We found that the absence of Rip11 resulted in a significant reduction in the export of typhoid toxin to the extracellular medium relative to parental cell (*Figure 5d* and *Figure 5—source data 1*), despite equivalent levels of typhoid toxin expression (*Figure 5e*) and typhoid toxin carrier formation (*Figure 5f*, *Figure 5—figure supplement 2*, *Figure 5—figure supplement 3* and *Figure 6—source data 1*), a phenotype similar to that observed for the absence of Rab11B (*Figure 5a–c* and *Figure 5—source data 1*). Taken together, our results suggest that Rab11B in concert with its effector protein Rip11 mediates steps in the transport of typhoid toxin-containing vesicle carriers most likely from the SCVs to the plasma membrane.

## VAMP7, SNAP23, and STX4 control the last steps of typhoid toxin export

The export of typhoid toxin is expected to involve steps in which the vesicle carriers harboring the toxin fuse with the plasma membrane and release their cargo to the extracellular space. In most exocytic pathways this step involves soluble *N*-ethylmaleimide-sensitive factor-activating protein receptors (SNAREs), which mediate the fusion between the carrier vesicle and target membranes

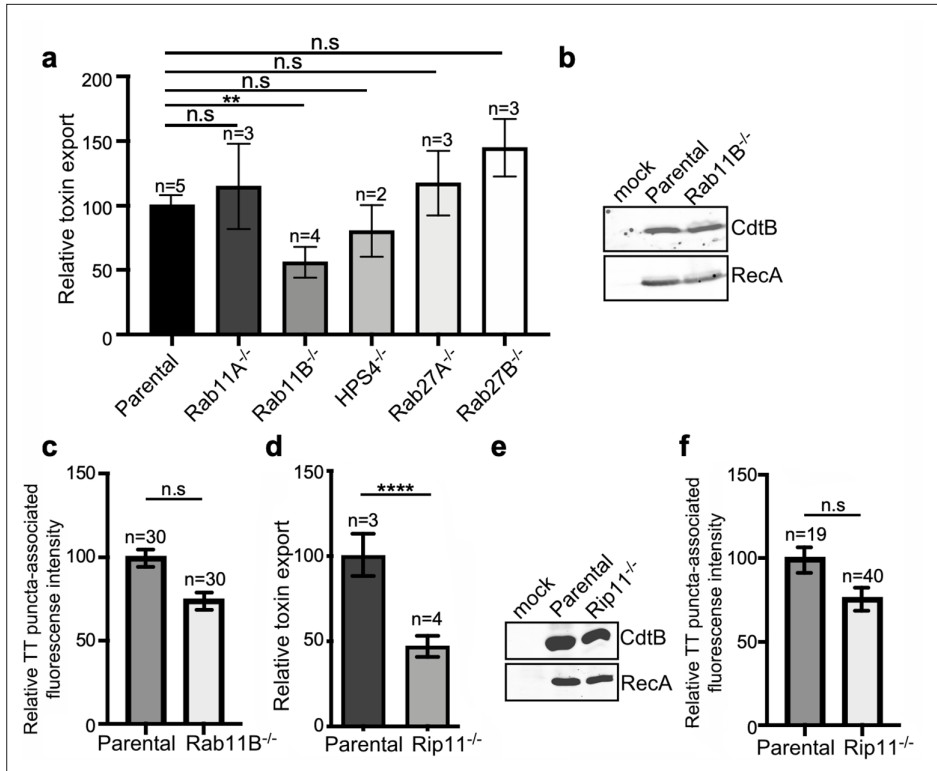

**Figure 5.** Rab11B and Rip11 regulate the transport of typhoid toxin vesicle carriers. (**a**) Typhoid toxin export into the infection medium. The quantification of the levels of typhoid toxin in the infection media was carried out by serial dilutions as indicated in the legend for **Figure 2**. Infection media obtained from *Salmonella* Typhi-infected HEK293T parental and the indicated deficient cells were serially diluted and applied to uninfected HEK293T cells. The cell cycle profile of treated cells was analyzed by flow cytometry, and the percentage of cells in the G2/M phase, a measure of typhoid toxin toxicity, was determined. The relative toxicity was determined by the percentage of cells at the G2/M phase after treatment with the different of the infection media fitted by nonlinear regression. Values were normalized relative to wild-type cells, which was considered to be 100 and are the mean ± SD of three independent experiments. **: p<0.01, unpaired two-sided *t* test. n.sd: differences not statistically significant. (**b**) Western blot analysis of the expression of typhoid toxin in parental HEK293T and Rab11B-deficient cells. Cells were infected with *S*. Typhi expressing FLAG-tagged CdtB, lysed 24 hr after infection and analyzed by western blot with antibodies directed to the FLAG epitope and the *Salmonella* protein RecA. (**c and f**) Quantification of the intensity of fluorescent puncta associated with typhoid toxin carrier intermediates in parental HEK293T and Rab11B- (**c**) or Rip11-deficient (**f**) cells. Values represent relative fluorescence intensity and are the mean ± SEM. n.s: differences not statistically significant. The results of an additional independent experiment are shown in **Figure 5—figure supplement 2**. (**e**) Western blot analysis of the expression of typhoid toxin in parental HEK293T and Rip11-deficient cells carried out as indicated in (**b**). (**d**) Relative typhoid toxin export in Rip11-deficient cells. Relative toxin export was determined as indicated in (**a**). Values were normalized relative to parental cells, which was considered to be 100, and are the mean ± SEM. ****: p<0.0001, unpaired two-sided *t* test. TT: typhoid toxin.

The online version of this article includes the following source data and figure supplement(s) for figure 5:

**Source data 1.** Raw data of **Figure 5a, c–d and f**.

**Source data 2.** Unprocessed CdtB and RecA western blots.

**Source data 3.** Unprocessed CdtB and RecA western blots.

**Figure supplement 1.** Genotyping of the indicated CRISPR/Cas9-edited cell lines.

**Figure supplement 1—source data 1.** Unprocessed DNA gels of genotyping of indicated cell lines.

**Figure supplement 1—source data 2.** Unprocessed DNA gels of genotyping of indicated cell lines.

**Figure supplement 2.** Analysis of independent clones of the CRISPR-Cas9-generated Rab27A-, Rab27B-, and Rip11-deficient cells shown in **Figure 5**.

**Figure supplement 3.** Rab11B and Rip11 regulate the transport of typhoid toxin vesicle carriers.

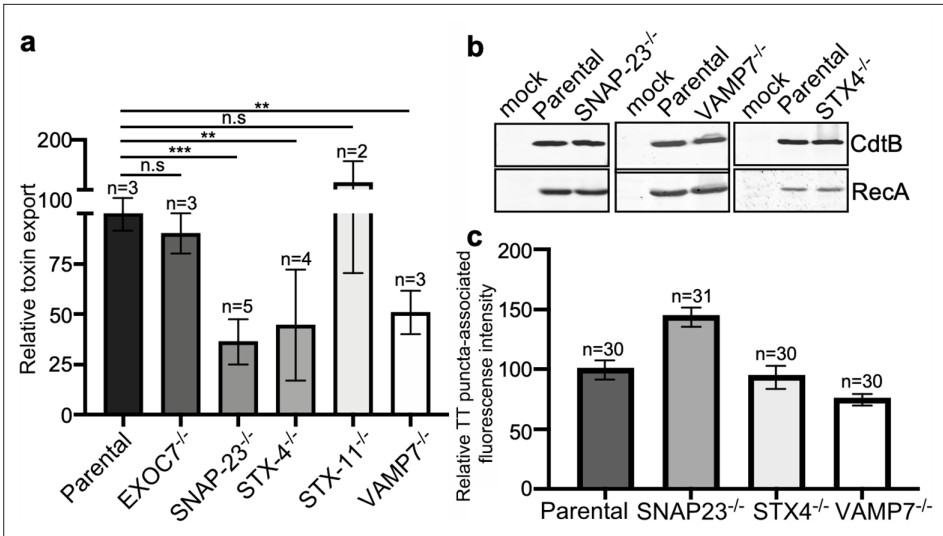

**Figure 6.** VAMP7, SNAP23, and STX4 control the last steps of typhoid toxin export. (**a**) Quantification of typhoid toxin export into the infection medium. Infection media obtained from *Salmonella* Typhi-infected HEK293T parental and the indicated deficient cells were serially diluted and applied to uninfected HEK293T cells. The cell cycle profile of treated cells was analyzed by flow cytometry, and the percentage of cells at the G2/M phase, a measure of typhoid toxin toxicity, was determined. The relative toxicity was determined by the percentage of cells at the G2/M phase from the dilution of infection media fitted by nonlinear regression. Values were normalized relative to parental cells, which was considered to be 100 and are the mean ± SD. ***: p<0.001; **: p<0.01, unpaired two-sided *t* test. (**b**) Western blot analysis of the expression of typhoid toxin in parental HEK293T and the SNAP-23-, VAMP7-, and STX4-deficient cells. Cells were infected with *S.* Typhi expressing FLAG-tagged CdtB, lysed after 24 hr of infection, and analyzed by western blot with antibodies directed to FLAG and the *Salmonella* protein RecA. (**c**) Quantification of the intensity of fluorescent puncta associated with typhoid toxin carrier intermediates in parental HEK293T and the indicated deficient cells. Cells were infected with a *S.* Typhi strain expressing FLAG-tagged CdtB and the levels of fluorescence associated with typhoid toxin carriers were determined 24 hr after infection. Values represent relative fluorescence intensity and are the mean ± SEM. n.s: differences not statistically significant. The results of an additional independent experiment are shown in *Figure 5—figure supplement 3*. TT: typhoid toxin.

The online version of this article includes the following source data and figure supplement(s) for figure 6:

**Source data 1.** Raw data of *Figure 6a and c*.

**Source data 2.** Unprocessed CdtB and RecA western blots.

**Figure supplement 1.** VAMP7, SNAP23, and STX4 control the last steps of typhoid toxin export.

**Figure supplement 2.** Analysis of independent clones of the CRISP-Cas9-generated EXOC7-, SNAP23-, STX4-STX11-, and VAMP7-deficient cells shown in *Figure 6*.

---

(*Goda, 1997*; *Südhof and Rothman, 2009*). Depending on whether they are located in the vesicle or target membranes, SNAREs are referred to as v- or t-SNAREs, respectively. The identities of the trans-SNARE pairing ensure the specificity of the fusion process. We therefore hypothesized that a pair of v- and t-SNAREs must be involved in targeting typhoid toxin-containing vesicles to the plasma membrane. We reasoned that a good candidate t-SNARE to be potentially involved in typhoid toxin export would be SNAP23, which is ubiquitously expressed, localizes to the plasma membrane, and has been implicated in several exocytic processes (*Kádková et al., 2019*). We therefore generated SNAP23-deficient cells by CRISPR/Cas9 gene editing (*Figure 5—figure supplement 1*), infected them with *S.* Typhi, and examined the levels of typhoid-toxin carrier intermediates in the infected cells and the levels of typhoid toxin in the culture media. We found that the amount of typhoid toxin in the infection media of SNAP23-deficient cells was significantly reduced when compared to the parental cell line (*Figure 6a* and *Figure 6—source data 1*), despite equivalent amount of toxin expression (*Figure 6b*). In contrast, we found that the levels of typhoid toxin vesicle carrier intermediates were actually increased in the SNAP23-deficient cells (*Figure 6c*, *Figure 6—figure supplement 1*, *Figure 6—figure supplement 2*, and *Figure 6—source data 1*). We hypothesize that this increase

is related to the failure of those carriers to fuse with the plasma membrane due to the absence of SNAP23. Taken together, these results are consistent with the hypothesis that SNAP23 is involved in the fusion of the typhoid toxin vesicle carrier intermediates with the plasma membrane.

SNAP23 has been shown to cooperate with other SNARE proteins such as syntaxin 4 (STX4) and syntaxin 11 (STX11) to mediate membrane fusion (*Lin et al., 2017*; *Ye et al., 2012*). Therefore, using CRISPR/Cas9 gene editing, we generated STX4- and STX11-deficient cells (*Figure 5—figure supplement 1*) and examined the export of typhoid toxin in the resulting cells after infection with *S.* Typhi. We found that the level of typhoid toxin in the infection media of STX4-deficient cells was significantly reduced in comparison to the parental cell line (*Figure 6a*) despite equivalent levels of toxin expression (*Figure 6b*). In contrast, typhoid toxin export was increased in STX11-deficient cells (*Figure 6a* and *Figure 6—figure supplement 2*). In addition, similar to SNAP23-deficient cells, we observed an increased amount of typhoid toxin carrier intermediates in the cytosol of STX4-deficient cells infected with *S.* Typhi (although the difference did not reach statistical significance), most likely due to the failure of these carriers to fuse with the plasma membrane (*Figure 6c*, *Figure 6—figure supplement 1*, and *Figure 6—figure supplement 2*, and *Figure 6—source data 1*).

The SNAP23/STX4 complex has been shown to interact with the v-SNARE VAMP-7 (*Röhl et al., 2019*; *Williams et al., 2014*). Therefore, in an attempt to identify the v-SNARE potentially responsible for the fusion of typhoid toxin carriers with the plasma membrane, we tested the export of typhoid toxin on VAMP-7-deficient cells generated by CRISPR/Cas9 gene editing (*Figure 5—figure supplement 1*). Consistent with the involvement of VAMP-7 in typhoid toxin export, we found reduced toxin levels in the infection media of VAMP-7-deficient cells infected with S. Typhi (*Figure 6a* and *Figure 6—source data 1*), although the levels of typhoid toxin carrier intermediates were not significantly altered in these cells (*Figure 6c* and *Figure 5—figure supplement 3*, and *Figure 6—figure supplement 2*). Collectively, these results indicate that the last step in typhoid toxin export is controlled by the VAMP7/SNAP23/STX4 SNARE complex, which targets the typhoid toxin vesicle carrier intermediates for fusion to the plasma membrane and subsequent release of the toxin cargo to the extracellular space.

## Discussion

Typhoid toxin has a very unusual biology for an exotoxin in that it is produced by intracellularly localized bacteria, is secreted into the lumen of the *S.* Typhi-containing vacuole, is packaged into vesicle intermediates, and then transported to the extracellular milieu (*Chang et al., 2019*; *Chang et al., 2016*; *Geiger et al., 2020*; *Geiger et al., 2018*; *Hodak and Galán, 2012*; *Spanò et al., 2008*). Here, we have investigated the mechanisms by which typhoid toxin is transported from the *S.* Typhi-containing vacuole to the extracellular space. Through an affinity purification approach, we identified CI-M6PR as the typhoid toxin packaging receptor, which we found to be robustly recruited to the *S.* Typhi-containing vacuole. This finding was surprising since it is well established that this receptor is excluded from the *S.* Typhimurium-containing vacuole (*Garcia-del Portillo and Finlay, 1995*; *McGourty et al., 2012*). Consistent with these observations, we found that typhoid toxin is not packaged into vesicle carrier intermediates when expressed in *S.* Typhimurium, indicating that the specific features of the *S.* Typhi-containing vacuole are essential for the formation of the typhoid toxin transport carrier, underlying the marked differences between the intracellular compartments that harbor these pathogens. The properties of the SCVs are determined by the activity of bacterial effectors delivered by either of its type III secretion systems, which differ significantly between *S.* Typhi and *S.* Typhimurium. In fact, we found that expression in *S.* Typhi of SseJ, a *S.* Typhimurium effector of its SPI-2 T3SS that is absent from *S.* Typhi (*Kolodziejek and Miller, 2015*; *Ohlson et al., 2005*), prevented the recruitment of CI-M6PR to the *S.* Typhi-containing vacuole and the subsequent packaging of typhoid toxin into vesicle carrier intermediates. How the presence of SseJ prevents the intersection of the *S.* Typhi-containing vacuole with CI-M6PR is not clear. It is known that through its acetyl transferase activity, SseJ modifies the lipid composition of the SCV membrane. This activity must be central for the exclusion of the CI-M6PR from the *S.* Typhi-containing vacuole since expression of a SseJ catalytic mutant did not affect the recruitment of CI-M6PR to the *S.* Typhi-containing vacuole, and did not interfere with typhoid toxin packaging. However, how modification of the lipid composition of the SCV membrane affects the trafficking of the SCV is unknown. These findings not only further highlight the differences between the intracellular compartments that harbor *S.* Typhi and

*S.* Typhimurium but also indicate that typhoid toxin has evolved to adjust its biology to the intracellular biology of *S.* Typhi.

The engagement of the sorting receptor by typhoid toxin presumably initiates a packaging and budding event that leads to the formation of the vesicle carrier intermediates. We found that this step requires the activities of the coat protein complex COPII and the Sar1 small GTPase. Cells defective in SEC23B or SAR1B showed reduced levels of typhoid toxin carriers and toxin export to the extracellular space despite indistinguishable levels in the number of intracellular *S.* Typhi and the levels of typhoid toxin expression. Furthermore, consistent with the involvement of the COPII coat in the formation of toxin carriers, typhoid toxin was shown to co-immunoprecipitate with SEC23B, a COPII coat component. Importantly, formation of the SEC23B-COPII complex was significantly impaired in the absence of CI-M6PR indicating that the sorting receptor serves as a bridge between the toxin cargo and the coat complex. COPII coats are involved in the early secretory pathway, more specifically in the ER to Golgi transport (*McCaughey and Stephens, 2018*) suggesting that *S.* Typhi-containing vacuole must intersect with elements of the secretory pathway.

Our studies also identified other components of the membrane trafficking machinery that are presumably involved in the transport of the vesicle carriers from the SCV to the plasma membrane. In particular, we identified Rab11B as required for the efficient transport of typhoid toxin to the extracellular space. Although cells deficient in Rab11B showed reduced amount of toxin in the infection media, they showed normal levels of vesicle carrier intermediates. These results indicate that Rab11B must be involved in steps downstream from the vesicle transport budding from the SCV but upstream from the fusion of these vesicles to the plasma membrane. Rab11B has been implicated in various vesicle transport pathways to the plasma membrane including melanin exocytosis (*Tarafder et al., 2014*). This observation is intriguing since the *S.* Typhi-containing vacuole (but not the S. Typhimurium vacuole) recruits Rab32 and Rab38, which are involved in melanosome biogenesis (*Spanò and Galán, 2012*; *Spanò et al., 2016*). However, we found that Rab32 and Rab38 are not required for typhoid toxin transport since cells deficient in HPS4, an essential component of their exchange factor (*Gerondopoulos et al., 2012*), were unaffected in typhoid toxin transport. The mechanisms by which Rab11B may regulate typhoid toxin transport are unclear. However, we hypothesize that this GTPase may be involved in the movement of the typhoid toxin carriers along microtubule tracks. This hypothesis is based on the observation that Rab11B has been shown to modulate the exocytic transport of insulin granules at least in part by linking these transport carriers to kinesin motors, a function that requires the Rab11B effector Rip11 (*Sugawara et al., 2009*). Consistent with this hypothesis, cells lacking Rip11 exhibited reduced transport of typhoid toxin to the extracellular space although they showed normal levels of typhoid toxin transport carriers.

Finally, we identified the machinery that is required for the fusion of the typhoid toxin transport carriers to the plasma membrane. More specifically we found that the plasma membrane SNARE proteins SNAP23 and syntaxin 4 are required for typhoid toxin transport to the extracellular space. Consistent with their involvement in typhoid toxin transport, cells lacking SNAP23 or syntaxin 4 showed markedly reduced levels of typhoid toxin in the infection media. Importantly, consistent with their specific involvement in the fusion of the typhoid toxin vesicle carriers with the plasma membrane, SNAP23- or syntaxin 4-deficient cells showed increased levels of typhoid toxin carrier in *S.* Typhi-infected cells. The SNAP23/STX4 complex has been shown to interact with the v-SNARE VAMP-7 (*Röhl et al., 2019*; *Williams et al., 2014*), and we found here that cells defective in VAMP-7 showed reduced levels of typhoid toxin in the extracellular media. Therefore, we propose that the fusion of the typhoid toxin vesicle carrier intermediates with plasma membrane is controlled by the plasma membrane t-SNAREs SNAP23 and STX4, and the typhoid toxin vesicle carrier v-SNARE VAMP7.

There are some limitations to this study. For example, we have identified the CI-M6PR as the typhoid toxin sorting receptor, which the toxin engages through interactions with Neu5Ac-terminated sialoglycans on the receptor. It remains possible that in other cells, typhoid toxin sorting may be mediated by other Neu5AC-bearing receptors. However, the ubiquitous distribution of CI-M6PR suggests that this receptor is likely to play a major role in all cells. Another limitation to this study is that it was not comprehensive since the nature of the experimental system did not allow us to conduct a genome-wide screen for cellular factors involved in typhoid toxin export, which could have identified additional components. We have previously utilized such an approach for the study of the typhoid toxin incoming transport pathway (*Chang et al., 2019*). Finally, like any approach relying on

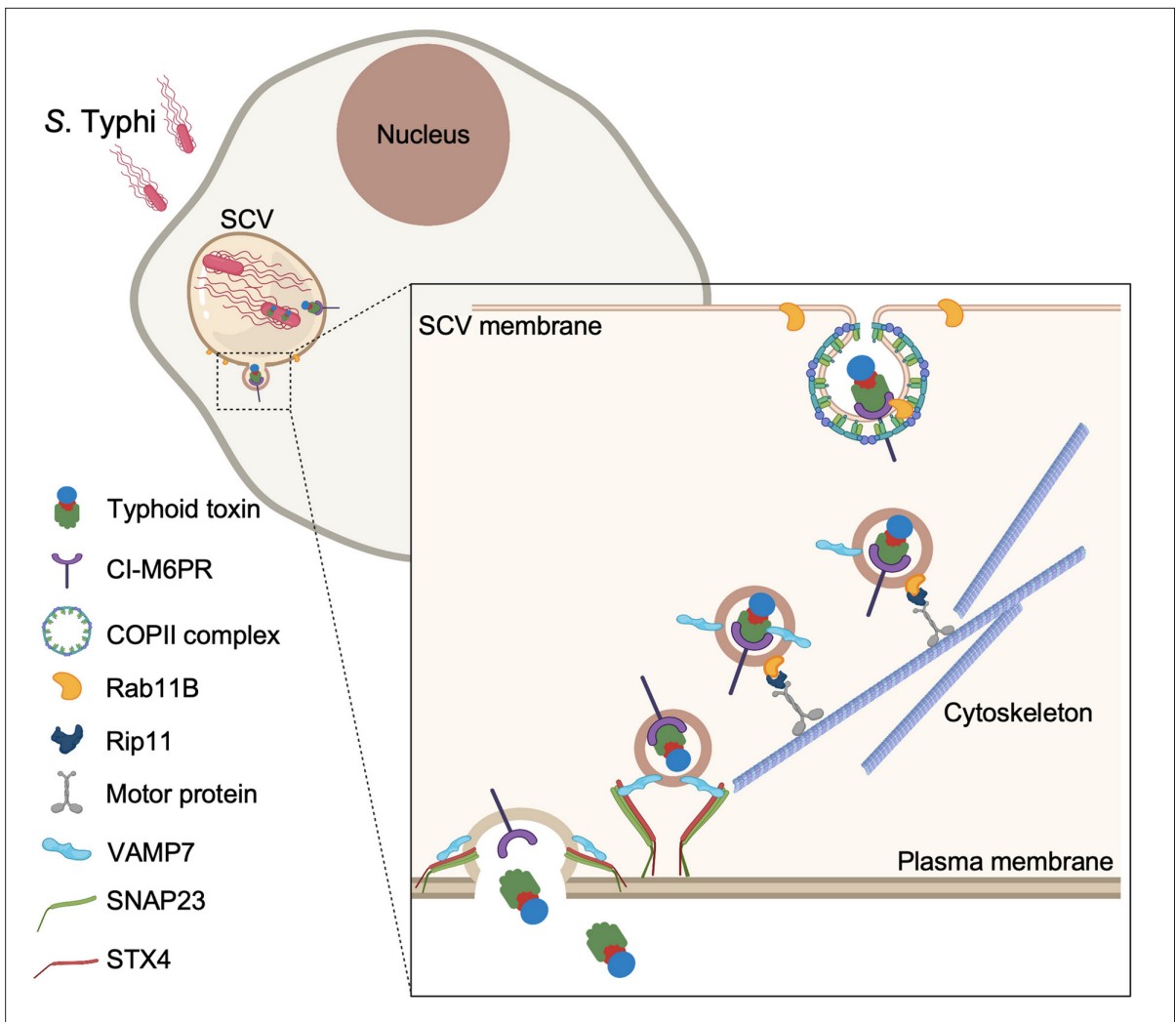

**Figure 7.** Model for typhoid toxin export from *Salmonella* Typhi-infected cells. Mediated by the action of effector proteins of its two type III secretion systems, *Salmonella* Typhi gains access to host cells and builds an intracellular niche that allows its replication and survival. Within this intracellular niche, *S.* Typhi expresses typhoid toxin, which is secreted to the lumen of the *Salmonella*-containing vacuole by a type X secretion system. Though interaction with its glycan-receptor-binding B subunit, typhoid toxin engages cation-independent mannose-6-phosphate receptor (CI-M6PR), which acts as its sorting receptor and mediates its packaging into vesicle transport carriers aided by COPII complex and Rab11b. The vesicle carriers are transported to the cell periphery by motor proteins engaging the Rab11b-interacting protein Rip11. On the cell periphery, the vesicle carriers undergo exocytosis by fusing to the plasma membrane, a process mediated by SNARE proteins VAMP7 on the vesicle carriers and SNAP23 and STX4 on the plasma membrane.

the introduction of mutations to study a phenotype, our study is vulnerable to effects caused by the mutations that may indirectly affect the phenotype under study. For example, as COPII is involved in export pathways from the endoplasmic reticulum, its mutation may result in off-target effects. Whenever possible, however, we have attempted to provide alternative pieces of evidence to support our conclusions.

In summary, our studies have unraveled the mechanisms by which typhoid toxin is transported from the SCV to the extracellular space. The export pathway, which has evolved to be specifically adapted to the biology of *Salmonella* Typhi, has coopted cellular machinery involved in various secretory and exocytic pathways (*Figure 7*). These findings reveal how vesicle trafficking pathways that are seemingly unconnected can be coopted by microbial pathogens to carry out a specific function.

# Materials and methods

**Key resources table**

| Reagent type (species) or resource | Designation | Source or reference | Identifiers | Additional information |
|---|---|---|---|---|
| Gene (*Salmonella enterica* serovar Typhi) | *cdtB* | PMID:18191792; MicrobesOnline Database | STY1886 | |
| Gene (*S.* Typhi) | *pltA* | PMID:18191792; MicrobesOnline Database | STY1890 | |
| Gene (*S.* Typhi) | *pltB* | PMID:18191792; MicrobesOnline Database | STY1891 | |
| Gene (*Salmonella enterica* serovar Typhimurium) | *SopD2* | MicrobesOnline Database | STM0972 | |
| Gene (*Salmonella enterica* serovar Typhimurium) | *GtgE* | MicrobesOnline Database | STM1055 | |
| Gene (*Salmonella enterica* serovar Typhimurium) | *SseJ* | MicrobesOnline Database | STM1631 | |
| Gene (*Salmonella enterica* serovar Typhimurium) | *SteB* | MicrobesOnline Database | STM14_1970 | |
| Gene (*Salmonella enterica* serovar Typhimurium) | *SlrP* | MicrobesOnline Database | STM0800 | |
| Gene (*Salmonella enterica* serovar Typhimurium) | *SseK1* | MicrobesOnline Database | STM4157 | |
| Gene (*Salmonella enterica* serovar Typhimurium) | *SseK2* | MicrobesOnline Database | STM2137 | |
| Gene (*Salmonella enterica* serovar Typhimurium) | *SseK3* | MicrobesOnline Database | STM14_2428 | |
| Gene (*Salmonella enterica* serovar Typhimurium) | *GtgA* | MicrobesOnline Database | STM14_1166 | |
| Gene (*Salmonella enterica* serovar Typhimurium) | *SseL* | MicrobesOnline Database | STM2287 | |
| Gene (*Salmonella enterica* serovar Typhimurium) | *SpvC* | KEGG Database | STM14_5561 | |
| Gene (*Salmonella enterica* serovar Typhimurium) | *SpvD* | KEGG Database | STM14_5560 | |
| Gene (*Salmonella enterica* serovar Typhimurium) | *GogB* | MicrobesOnline Database | STM2584 | |
| Gene (*Salmonella enterica* serovar Typhimurium) | *SspH1* | MicrobesOnline Database | STM14_1483 | |
| Cell line (*Homo sapiens*) | Henle-407 | Roy Curtiss laboratory collection | | Cell line maintained in Jorge Galan's lab |
| Cell line (*Homo sapiens*) | HEK293T | American Type Culture Collection (ATCC) | | Cell line maintained in Jorge Galan's lab |
| Strain, strain background (*Salmonella enterica* serovar Typhi) | ISP2825 | PMID:1879916 | | |
| Strain, strain background (*Salmonella enterica* serovar *Typhimurium*) | SL1344 | PMID:1879916 | | |
| Recombinant DNA reagent | WT typhoid toxin; pSB5496 | This paper | | Vector(pET28b) to express pltA-pltB-cdtB-3xflag-His in the *E. coli* BL21 cells |
| Recombinant DNA reagent | Mutant typhoid toxin (PltB[S35A] toxin); pSB5497 | This paper | | Vector(pET28b) to express pltA-pltB S35A-cdtB-3xflag-His in the *E. coli* BL21 cells |
| Recombinant DNA reagent | pWSK-SopD2 | PMID:28742135 | STM0972 | |
| Recombinant DNA reagent | pWSK-GtgE | PMID:28742135 | STM1055 | |
| Recombinant DNA reagent | pWSK-SseJ | PMID:28742135 | STM1631 | |
| Recombinant DNA reagent | pWSK-SteB | PMID:28742135 | STM14_1970 | |
| Recombinant DNA reagent | pWSK-SlrP | PMID:28742135 | STM0800 | |

| Reagent type (species) or resource | Designation | Source or reference | Identifiers | Additional information |
|---|---|---|---|---|
| Recombinant DNA reagent | pWSK-SseK1 | PMID:28742135 | STM4157 | |
| Recombinant DNA reagent | pWSK-SseK2 | PMID:28742135 | STM2137 | |
| Recombinant DNA reagent | pWSK-SseK3 | PMID:28742135 | STM14_2428 | |
| Recombinant DNA reagent | pWSK-GtgA | PMID:28742135 | STM14_1166 | |
| Recombinant DNA reagent | pWSK-SseL | PMID:28742135 | STM2287 | |
| Recombinant DNA reagent | pWSK-SpvC | PMID:28742135 | STM14_5561 | |
| Recombinant DNA reagent | pWSK-SpvD | PMID:28742135 | STM14_5560 | |
| Recombinant DNA reagent | pWSK-GogB | PMID:28742135 | STM2584 | |
| Recombinant DNA reagent | pWSK-SspH1 | PMID:28742135 | STM14_1483 | |
| Recombinant DNA reagent | pWSK-SseJ$^{S151A}$; pSB5848 | This paper | | The plasmid to express mutant SseJ$^{S151A}$ in *Salmonella* Typhi |
| Recombinant DNA reagent | pEGFP-Sec23A | Addgene; PMID:10825291 | | |
| Recombinant DNA reagent | px459-CI-M6PR | This paper | sgRNA | gRNA sequence: CGGACTGAAGCTGGTGCGCA |
| Recombinant DNA reagent | px459-Sec23B | This paper | sgRNA | gRNA sequence: TACAATTGAGTACGTGATAC |
| Recombinant DNA reagent | px459-AP4m1 | This paper | sgRNA | gRNA sequence: CTCTTTGACCTCAGCAGCGT |
| Recombinant DNA reagent | px459-CLTC | This paper | sgRNA | gRNA sequence: TCGTTTTCAGGAGCATCTCC |
| Recombinant DNA reagent | px459-AP3B1 | This paper | sgRNA | gRNA sequence: AAAGAAGAAGCCGTATACTA |
| Recombinant DNA reagent | px459-Sar1B | This paper | sgRNA | gRNA sequence: AATGTGCCTATACTGATTCT |
| Recombinant DNA reagent | px459-Rab11A | This paper | sgRNA | gRNA sequence: CATTTCGAGTAAATCGAGAC |
| Recombinant DNA reagent | px459-Rab11B | This paper | sgRNA | gRNA sequence: GAGCAAGAGCACCATCGGCG |
| Recombinant DNA reagent | px459-HPS4 | This paper | sgRNA | gRNA sequence: GAAGGCGATCCAACAAGAGC |
| Recombinant DNA reagent | px459-Rab27A | This paper | sgRNA | gRNA sequence: AGTGGCTCCATCCGGCCCAC |
| Recombinant DNA reagent | px459-Rab27B | This paper | sgRNA | gRNA sequence: TCCTGGCCCTCGGGGATTCA |
| Recombinant DNA reagent | px459-Rip11 | This paper | sgRNA | gRNA sequence: GGTCAAACATACTGGCGCTC |
| Recombinant DNA reagent | px459-Exoc7 | This paper | sgRNA | gRNA sequence: TGACGAAGGCACTGACGCAG |
| Recombinant DNA reagent | px459-SNAP23 | This paper | sgRNA | gRNA sequence: AAGACAACATGGGGAGATGG |
| Recombinant DNA reagent | px459-STX-4 | This paper | sgRNA | gRNA sequence: TGGTGCACCCGGGCACGGCA |
| Recombinant DNA reagent | px459-STX-11 | Other | | A gift from Dr Craig Roy's lab |
| Recombinant DNA reagent | px459-VAMP7 | This paper | sgRNA | gRNA sequence: TTCTGAATGAGATAAAGAAG |
| Antibody | Anti-FLAG M2 (mouse monoclonal) | Sigma-Aldrich | SI-F3165-1MG | IF (1: 5000) WB (1:10,000) |
| Antibody | Anti-*S*. Typhi (rabbit polyclonal) | Sifin | TS1605 | IF (1:10,000) |
| Antibody | Anti-IGF-II Receptor/CI-M6PR (D3V8C) (rabbit monoclonal) | Cell Signaling Technology (CST) | #14364 | IF (1:500) WB (1:500) |
| Antibody | Anti-GFP (rabbit polyclonal) | GeneTex | GTX113617 | WB (1:10,000) |
| Antibody | Anti-GFP (mouse monoclonal) | GeneTex | GTX628528 | WB (1:10,000) |
| Antibody | Goat anti-mouse IgG1 Alexa-Fluor 488 (goat polyclonal) | Thermo Fisher Scientific | A21121 | IF (1:2000) |
| Antibody | Goat anti-rabbit IgG (H+L) Alexa-Fluor 594 (goat polyclonal) | Thermo Fisher Scientific | A11012 | IF (1:2000) |
| Software, algorithm | ImageJ with Microbe J plug-in | http://rsbweb.nih.gov/ij/; PMID:27572972 | | |
| Software, algorithm | R software | https://www.r-project.org/; PMID:30357717 | | |

| Reagent type (species) or resource | Designation | Source or reference | Identifiers | Additional information |
|---|---|---|---|---|
| Software, algorithm | Imaris | https://imaris.oxinst.com/ | | |

## Bacterial strains and plasmids

The wild-type *S. enterica* serovars Typhi ISP2825 (*Galán and Curtiss, 1991*) and Typhimurium SL1344 (*Hoiseth and Stocker, 1981*) have been described previously. All the *Salmonella* mutant derivatives were constructed by standard recombinant DNA and allelic exchange procedures as previously described (*Kaniga et al., 1994*) using the *Escherichia coli* β-2163 *Δnic35* strain (*Demarre et al., 2005*) as the conjugative donor and are listed in the Key resources table. All the plasmids listed in the Key resources table were constructed using the Gibson assembly cloning strategy (*Gibson et al., 2009*) and verified by nucleotide sequencing.

## Antibodies

Antibodies to CI-M6PR (Cell Signaling Technology, Cat. #14364), anti-FLAG M2 (Sigma, Cat.# F1804), and to *Salmonella* LPS (Sifin) were purchased from the indicated commercial sources.

## Cell culture and bacterial infection

Human intestinal epithelial Henle-407 cells (obtained from the Roy Curtiss III collection in 1987) and HEK293T cells (from the American Type Culture Collection) were cultured in DMEM supplemented with 10% fetal bovine serum. Overnight cultures of the *S.* Typhi strains were diluted 1/20 in LB broth medium containing 0.3 M NaCl and grown until they reached an OD600 of 0.9. Culture cells were infected as described before (*Chang et al., 2016*). Briefly, cells were infected with the different *S.* Typhi or *S.* Typhimurium strains at a multiplicity of infections (MOI) of 30 or 10, respectively, in Hank's balanced salt solution, and then incubated with culture medium containing 100 µg/ml gentamicin for 1 hr to kill extracellular bacteria. Cells were washed and incubated for the indicated times with medium containing 10 µg/ml gentamicin to avoid cycles of reinfection. All cell lines were routinely tested for a mycoplasma by a standard PCR method.

## Visualization of typhoid toxin export carrier intermediates

The visualization of typhoid toxin vesicle carrier intermediates was carried out as previously described (*Chang et al., 2016*). Briefly, 24 hr after infection with the indicated *S.* Typhi strains expressing FLAG-epitope-tagged CdtB, cells were fixed in 4% paraformaldehyde and then blocked with 3% BSA, 0.3% Triton X-100 in DPBS. Fixed cells were incubated with primary mouse monoclonal anti-FLAG M2 (Sigma) and rabbit polyclonal anti-*S.* Typhi lipopolysaccharide (Sifin) antibodies followed by Alexa 488-conjugated anti-mouse and Alexa 594-conjugated anti-rabbit antibodies (Invitrogen). Samples were visualized under a Nikon TE2000 fluorescence microscope equipped with and Andor Zyla 5.5 CMOS camera driven by Micromanager software (https://www.micro-manager.org). Quantification of typhoid toxin export carrier intermediates has been previously described (*Chang et al., 2016*). Briefly, images were analyzed using the open-source software ImageJ (https://imagej.nih.gov/). The LPS stain was used to identify the area corresponding to the bacterial cell body and this area was used to obtain the bacterial-associated typhoid toxin fluorescence signal, which was substracted from the typhoid toxin-associated fluorescence. The remaining fluorescence was defined as the area of typhoid toxin carrier intermediates. The intensity of fluorescence associated with toxin carriers was normalized using the fluorescence-associated typhoid toxin within bacterial cells in the same field.

## Immunofluorescent staining of CI-M6PR in infected cells

Henle-407 cells infected with the different *S.* Typhi or *S.* Typhimurium strains expressing superfolder GFP were fixed with ice-cold methanol for 5 min at –20°C. Cells were then washed with DPBS three times, permeabilized with 0.1% Triton-100/DPBS for 20 min at room temperature, and then stained with the antibodies against CI-M6PR and GFP overnight at 4°C, and Alexa 488 and 594-conjugated antibodies (Invitrogen) for 1 hr at room temperature. Cells were then observed under a Nikon TE2000 fluorescence microscope and images captured with an Andor Zyla 5.5 CMOS camera driven by Micromanager software (https://www.micro-manager.org). Co-localization between CI-M6PR and SCV were analyzed by performing the plug-in Manders' overlap coefficient in ImageJ.

## CRISPR/Cas9 gene inactivation

CRISPR/Cas9 genome editing was carried out as described previously (*Chang et al., 2016*). The sgRNA sequences for the generation of CRISPR/Cas9-edited cell lines used in this study, which are listed in the Key resources table, were cloned into plasmid pSpCas9(BB)-2A-Puro(PX459) V2.0 (*Ran et al., 2013*), which was a gift from Feng Zhang (Addgene plasmid # 62988; RRID:Addgene # 62988; http://n2t.net/addgene:62988). PX459 carrying sgRNA were transfected into HEK293T cells using Lipofectamine 2000. The transfected cells were selected in culture medium containing puromycin for 2 days, and isolated clones were screened by PCR genotyping to identify knockout cells. At least two independently isolated clones per cell line were examined for the relevant phenotypes.

## Bacteria internalization and intracellular replication

The gentamicin protection assay was used to assess the invasion of *S*. Typhi within host cells (*Galán and Curtiss, 1989*). Briefly, HEK293T cells were cultured in 24-well plates and infected with the indicated *Salmonella* strains. One hour after infection, cells were incubated with culture medium containing gentamicin and then harvested in DPBS containing 0.1% sodium deoxycholate at 1 and 24 hr after addition of the antibiotic. The bacteria released from cultured cells were plated onto LB agar plates to determine colony-forming units.

## Affinity purification-mass spectrometry analysis

HEK293T and Henle-407 cells were seeded onto each five 10 cm dishes at 70% confluency. After 16–18 hr, culture cells were harvested, lysed in 5 ml of lysis buffer (0.5% Triton X-100, 150 mM NaCl, 50 mM Tris-HCl) containing protease inhibitors for 30 min on ice, and then centrifuged for 15 min at 14,000×rpm at 4°C. The supernatants were incubated with 30 µg of the purified typhoid toxin overnight at 4°C. Next day, the protein complex was incubated with 20 µl of anti-FLAG M2 agarose for 2 hr at 4°C. Immunoprecipitation of protein complexes were collected by centrifugation at 500× *g* for 1 min, followed by washing with lysis buffer to avoid non-specific binding. The protein complexes were eluted by adding 0.1 M glycine HCl, pH 3.5, and digested in solution with trypsin overnight. The extracted peptides were subjected to LC-MS/MS analysis as previously described (*Sun et al., 2018*).

## Co-immunoprecipitation assay

HEK293T cells were seeded at a density of $2 \times 10^6$ onto 10 cm dishes. Twenty-four hours later, cells were infected with *S*. Typhi strains expressing FLAG-epitope-tagged CdtB or mock-infected for 1 hr at an MOI of 10. Infected cells were harvested at 24 hr post infection and lysed in lysis buffer (0.5% Triton X-100, 150 mM NaCl, 50 mM Tris-HCl) containing protease inhibitors (cOmpleteTM, EDTA-free protease inhibitor cocktail, Roche). After 30 min on ice, samples were centrifuged for 15 min at 14,000×rpm at 4°C. The supernatants were incubated with 20 µl of anti-FLAG M2 agarose for 24 hr at 4°C. Protein complexes were collected by centrifugation at 500× *g* for 1 min, washed three times with lysis buffer, and then resuspended in 30 µl of Laemmli sample buffer. Samples were analyzed by SDS-PAGE and western blot with antibodies against FLAG and CI-M6PR.

## Typhoid toxin export assay

Cultured cells were grown onto six-well plates and infected with wild-type *S*. Typhi. Twenty-four hours after infection, the supernatant from infected cells was collected and filtered through 0.2 µm syringe filters, diluted as indicated in each experiment, and applied to fresh wild-type HEK293T cells. The toxic effect of different dilutions of the culture supernatant was determined at 48 hr post treatment as previously described (*Chang et al., 2016*). Briefly, treated cells were trypsinized, fixed for 1 hr in 70% ethanol/DPBS at –20°C, washed with DPBS, and then resuspend in DPBS containing 50 µg/ml propidium iodide, 0.1 mg/ml RNase A, and 0.05% Triton X-100. Cell cycle analysis was carried out by flow cytometry, and the proportion of cells in the G2/M phase was determined using FlowJo. Values were fitted to an orthogonal polynomial regression of degree 2 to estimate the relationship between different dilutions and 50% of cells in the G2/M phase using R software version 3.4.4 (https://www.r-project.org). Values analyzed from the R program were normalized relative to those of the parental cells, which were considered 100 and are the mean ± SD of three independent experiments.

## Toxin expression and purification

Purification of typhoid toxin was conducted as described previously (*Song et al., 2013*). Briefly, the genes encoding different versions of typhoid toxins in *S.* Typhi (as listed in the Key resources table) were cloned into the pET28a (Novagen) expression vector. *E. coli* strains carrying the plasmids encoding the different toxins were grown at 37°C in LB media to an OD600 of ~0.6, toxin expression was induced by the addition of 0.5 mM IPTG, and cultures were further incubated at 25° C overnight. Bacterial cell pellets were resuspended in a buffer containing 15 mM Tris-HCl (pH 8.0), 150 mM NaCl, 0.1 mg/ml DNase I, 0.1 mg/ml lysozyme, and 0.1% PMSF and lysed by passaging through a cell disruptor (Constant Systems Ltd.). Toxins were then purified from bacterial cell lysates through affinity chromatography on a Nickel-resin (Qiagen), ion exchange, and gel filtration (Superdex 200) chromatography as previously described (*Song et al., 2013*). Purified toxins were examined for purity on SDS-PAGE gels stained with coomassie blue.

## Statistical analysis

The p values were calculated using a two-tailed, unpaired Student's t-test for two-group comparisons in GraphPad Prism (GraphPad software); p values below 0.05 were considered statistically significant.

## Data and software availability

The following software was used in this study: GraphPad Prism (plotting data), Micro-Manager, Slidebook 6, Imaris 9.7 (Oxford Instruments), Adobe Illustrator & Adobe Photoshop (image preparation), FlowJo (analysis of flow cytometry data), and Image Studio Lite (Li-COR Biosciences), and Image Lab (Bio-Rad Laboratories, Inc) (quantification of the band intensity of western blot).

## Additional information

### Funding

| Funder | Grant reference number | Author |
|---|---|---|
| National Institute of Allergy and Infectious Diseases | AI079022 | Jorge E Galan |
| Ministry of Science and Technology, Taiwan | 110-2636-B-002 -014 - | Shu-Jung Chang |

The funders had no role in study design, data collection and interpretation, or the decision to submit the work for publication.

### Author contributions

Shu-Jung Chang, Conceptualization, Data curation, Formal analysis, Investigation, Methodology, Writing - review and editing; Yu-Ting Hsu, Yun Chen, Yen-Yi Lin, Investigation; Maria Lara-Tejero, Formal analysis, Investigation, Methodology, Writing - review and editing; Jorge E Galan, Conceptualization, Formal analysis, Funding acquisition, Methodology, Project administration, Supervision, Writing - original draft, Writing - review and editing

### Author ORCIDs

Shu-Jung Chang http://orcid.org/0000-0002-6037-6339
Maria Lara-Tejero http://orcid.org/0000-0002-1339-0859
Jorge E Galan http://orcid.org/0000-0002-6531-0355

### Decision letter and Author response

Decision letter https://doi.org/10.7554/eLife.78561.sa1
Author response https://doi.org/10.7554/eLife.78561.sa2

# Additional files

### Supplementary files
- Transparent reporting form
- Supplementary file 1. List of strains used in this study.
- Supplementary file 2. List of plasmids used in this study.
- Source data 1. Unprocessed western blots.

### Data availability
All data generated or analysed during this study are included in the manuscript and supporting files; source data files for all figures have been provided.

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
