## [Editor Report]

This paper is of interest to microbiologists as well as eukaryotic cell biologists interested in vesicular trafficking pathways. The authors identify several eukaryotic proteins required for typhoid toxin export from *Salmonella* Typhi-infected cells to the extracellular space including mannose-6-phosphate Receptor that serves sorting receptor.

---

## [Decision Letter]

**Decision letter after peer review:**

[Editors’ note: the authors submitted for reconsideration following the decision after peer review. What follows is the decision letter after the first round of review.]

Thank you for submitting the paper "Typhoid toxin sorting and exocytic transport from *Salmonella* Typhi infected cells" for consideration by *eLife*. Your article has been reviewed by 3 peer reviewers, and the evaluation has been overseen by a Reviewing Editor and a Senior Editor. The following individual involved in review of your submission has agreed to reveal their identity: Ethel Bayer-Santos (Reviewer #3).

Comments to the Authors:

We are sorry to say that, after consultation with the reviewers, we have decided that this work needs a major revision prior to be be considered for publication by *eLife*.

Despite being generally positive about the nature of this study, the reviewers have raised serious issues with this paper that they feel will take some time to address. While I am rejecting the paper as a result of this, I am broadly supportive of this manuscript and if you feel that you are able to address these issues, I will consider a newly submitted form of this paper that I will treat as a revised manuscript.

Among the shortcomings identified, the four points listed below would need to be addressed experimentally:

1. In order for you to conclude that COPII is required for SCV-plasma membrane trafficking they would have to show that 1) knockdown of sec23 does not have an indirect effect on M6PR by blocking its ER export – you could do this by examining the localization of M6PR at the TGN and endosomes by immunofluorescence microscopy and 2) COPII can mediate trafficking from the SCV You could do this by checking for co-localization of COPII components with the SCV or the typhoid toxin.

2. You should control for the possibility that secreted toxin undergoes proteolysis in cells lacking M6PR (or other proteins). If this is the case, then important conclusions might be unjustified.

3. You should complement the CRISPR KO cell lines described in this study with a plasmid expressing the deleted gene to confirm that the phenotype is not due to a random mutation acquired during the process. Alternatively, you could show same phenotype for independently generated mutants

4. The interaction between CdtB and Sec23 should be tested as control in the m6pr knockout background.

*Reviewer #1:*

The typhoid toxin is produced by *Salmonella* Typhi, the causative agent of typhoid fever in humans. The toxin is secreted into the *Salmonella*-containing vacuole (SCV) of host cells, from where it must be transported to the extracellular space to carry out its action. In this manuscript by Chang et al. the authors investigate the pathway by which the typhoid toxin is trafficked out of the host cell. They use immunoprecipitation followed by mass spectrometry to identify the cation-independent mannose-6-phosphate (CI-M6PR) as a binding partner for the typhoid toxin and convincingly demonstrate its role as the typhoid toxin sorting receptor. In addition, they show that the Typhimurium effector SseJ blocks recruitment of CI-M6PR to the SCV, confirming that the expression of specific Type III secretion system effectors affects the environment of the SCV. They then use CRISPR/cas9 knockdowns to characterise other host trafficking proteins that are required for toxin export, including COPII, Sar1, Rab11B, Rip11, VAMP7, SNAP23 and Syntaxin 4. These data provide interesting insights into the requirements of host proteins for typhoid toxin export, but a single pathway is not convincingly established, as proteins identified have not been linked together in terms of interaction or dependency with respect to toxin secretion. In addition, the data need to be discussed in the context of established knowledge of eukaryotic trafficking pathways.

1. The results of this work should be discussed in the context of the large body of published work on intracellular trafficking. M6PR intracellular trafficking pathways have been well characterised with information on the signalling sequences found in its cytoplasmic tail, the adaptor proteins they interact with including AP1, AP2, GGAs, PACS^-1^, TIP47 and Rab9 and the specific steps of trafficking pathways that they are involved in. The authors should discuss how their data relates to these pathways. AP1 is known to interact with M6PR and to traffic to and from endosomes and should therefore be included in the screens or at least discussed with a clear reason for its omission. The authors should also comment on the fact that the CI-M6PR normally traffics from the TGN to both endosomes and the plasma membrane, which could provide a plausible pathway.

2. COPII mediates trafficking of proteins between the ER and the Golgi. As such its knockdown should block all export from the ER including that of M6PR. The authors should discuss their data in the context of this information. To be convincing the authors need to rule out any indirect effect caused by blocking ER export and demonstrate clearly that COPII mediates this atypical pathway. For example, it is important to know (1) if COPII is recruited to the SCV membrane in a toxin-dependent manner, (2) if COPII interacts with M6PR and (3) is such an interaction required for toxin export in a way that can be distinguished from general ER export?

3. In order to convincingly establish a pathway, the proteins identified should be linked experimentally. For example, does the interaction of the toxin with Sec23 depend on the presence of M6PR? Can the authors be certain that all identified proteins are involved in direct traffic from the SCV to the plasma membrane? AP4 knockdown also decreased toxin export. Can the authors rule out its involvement in a separate step of the pathway?

4. The relative toxin export assay is not a convincing measurement. *Salmonella* infection and perturbation of mannose-6-phosphate receptor traffic have been shown to increase the secretion of proteases (PMID 32514074, 23162002), which could degrade extracellular secreted toxin. Why do the authors use an indirect assay for cell cycle arrest rather than western blot of supernatants? Western blot analysis should be used to determine if lower amounts in extracellular fractions can be rescued by protease inhibition. Can the authors discuss the relevance of an increased relative toxin export as seen after knockdown of AP3B1 and STX-11?

5. The screen of Typhimurium effectors is interesting – in relation to Typhimurium. But the results do not help understand the features of the Typhi SCV that result in CI-M6PR recruitment. Since the Typhi spiA mutant is defective in packaging toxin into transport carriers (Chang et al. 2016), is this mutant defective in recruiting CI-M6PR to the vacuole? Or is CI-M6PR recruitment simply part of a default phago-lysosomal pathway, as would occur with dead bacteria?

*Reviewer #2:*

Their data well justify the authors' conclusions. The article provides an excellent example highlighting the adaptation of an exotoxin co-evolved in the context of the life cycle of an intracellular pathogen producing the toxin. One remaining question not addressed in this article is whether CI-M6PR's role as a sorting receptor for typhoid toxin is shared across different host cells. This question arises because the interaction between CI-M6PR and typhoid toxin PltB is through the glycan, N-acetylneuraminic acid, displayed on the glycosylated sorting receptor(s), suggesting a different glycoprotein might serve as a sorting receptor in other host cells. Nonetheless, this manuscript elegantly determined the detailed sorting and transport process of typhoid toxin occurring in S. Typhi-infected host cells, which is well-justified by the data presented in the manuscript.

1. One remaining question not addressed in this article is whether CI-M6PR's role as a sorting receptor for typhoid toxin is shared across different host cells. This question arises because the interaction between CI-M6PR and typhoid toxin PltB is through the glycan, N-acetylneuraminic acid, displayed on the glycosylated sorting receptor(s), suggesting a different glycoprotein might serve as a sorting receptor in other host cells. This can be addressed by investigating other cells relevant to S. Typhi infection. Alternatively, the authors can address this point in the Discussion, perhaps under a separate section "Limitations of this study" to help readers obtain a clearer understanding of the toxin trafficking pathway.

2. SseJ's role is demonstrated, which is absent in S. Typhi. In some places in the manuscript (i.e., Abstract and Result, page 2, lines 11-13, page 7, lines 16-18), the corresponding descriptions are a bit confusing, sounding like a SPI2 effector produced by S. Typhi plays a key role in the packaging process. Rephrasing some of the descriptions may be considered to help readers get the concept quickly.

3. Relating to SseJ's role, examining the altered expression and/or localization of OSBP and CI-M6PR on the SCV in various cases tested in the manuscript (i.e., S. Typhi-, S. Typhi+sseJ-, and S. Typhi+sseJS151A-infected host cells) is expected to strengthen the authors' point on the co-evolution of SPI2 effectors and typhoid toxin.

4. S. Figures S1, S2, S4, S7, and S8: All data from 2-3 independent experiments can be combined and presented as the main figures (as opposed to one representative data for the main figures and all the remaining data for supplementary figures).

*Reviewer #3:*

In this manuscript by Chang et al., authors identified the intracellular receptor (cation-independent mannose-6-phosphate receptor, CI-M6PR) of *Salmonella* Typhi typhoid toxin , which is responsible for sorting this toxin out of the *Salmonella*-containing vacuole (SCV). This was a curious finding since it was well established by complementary studies using *Salmonella* Typhimurium that CI-M6PR does not colocalize with the SCV. The authors went on to determine the reason for such differences and identified the S. Typhimurium-specific T3SS effector SseJ as responsible for preventing the co-localization of CI-M6PR in SCVs.

The proteins playing a role in the exocytic pathway driving the export of typhoid toxin were also identified. After binding of the typhoid toxin to CI-M6PR in the lumen of the SCV, COPII and SarIB mediate the formation of export carries, which are transported to the plasma membrane in a manner dependent on the GTPase Rab11B and its interacting partner RIP11. Lastly, the fusion of the carriers to the plasma membrane depends on the SNARE proteins VAMP7, SNAP23 and STX4.

This work answers a long-standing question in the field, and the author conclusions are well supported by a series of complementary experiments.

Some controls are missing in this version of the manuscript, which could provide strength to these findings.

In my opinion, the work would greatly beneficiate from live-imaging microscopy analysis showing the trafficking of typhoid toxin within infected or transfected cells. These data could complement the current methodology and make the article more interesting.

1) I think that it is important to complement all the CRISPR KO cell lines described in this study with a plasmid expressing the deleted gene to confirm that the phenotype is not due to a random mutation acquired during the process (either due to the technique itself or the selection of clones within the initial WT population). As all data was performed in HEK293T cells that transfect very well, this could be done relatively easily for CI-M6PR-/-; SEC23B-/-; Sar1B-/-; Rab11B-/-; Rip11-/-; SNAP23-/-; STX4-/-; VAMP7-/-.

2) The co-IP experiment shown in Figure 4I could be performed in the CI-M6PR-/- background. If the authors hypothesis is correct, one would expect that the interaction between CdtB and Sec23 (COPII) would be lost.

3) Although discussed in the text, Figure 6C is lacking statistical analysis. This should be included.

[Editors’ note: further revisions were suggested prior to acceptance, as described below.]

Thank you for resubmitting your work entitled "Typhoid toxin sorting and exocytic transport from *Salmonella* Typhi infected cells" for further consideration by *eLife*. Your revised article has been evaluated by Dominique Soldati-Favre (Senior Editor) and a Reviewing Editor.

The manuscript has been improved but there are some remaining issues that need to be addressed, as outlined below:

They concur that the manuscript is significantly improved and recommended a few points to be addressed (that will not involve new experiments) prior to consideration for acceptance.

*Reviewer #1:*

In general, the authors have addressed the main concerns of the reviewers and the manuscript is much approved.

As COPII is responsible for ER export there is a high possibility that blocking it through knockout could result in the mislocalisation of many proteins and subsequent off-target effects; this is not fully addressed.

In addition, the authors do not discuss the extensive literature on M6PR trafficking including the role of AP1, nor do they conclusively rule out the role of AP1 in the trafficking of M6PR-bound toxin.

These points could be mentioned in the 'limitations of this study' paragraph in the Discussion.

*Reviewer #2:*

I consider that the authors have addressed my and other reviewers' comments in a satisfactory manner, and this new version of the manuscript is greatly improved.

*Reviewer #3:*

The revised version of the manuscript has been significantly improved. The authors have addressed all my concerns raised in the initial review.

---

## [Author Response]

[Editors’ note: the authors resubmitted a revised version of the paper for consideration. What follows is the authors’ response to the first round of review.]

Among the shortcomings identified, the four points listed below would need to be addressed experimentally:1. In order for you to conclude that COPII is required for SCV-plasma membrane trafficking they would have to show that 1) knockdown of sec23 does not have an indirect effect on M6PR by blocking its ER export – you could do this by examining the localization of M6PR at the TGN and endosomes by immunofluorescence microscopy.

In the original submission of the article, we stated that "Recruitment of the packaging receptor CI-M6PR to the S. Typhi-containing vacuoles in SEC23B-deficient cells was indistinguishable from the recruitment observed in the parental cells, indicating that the defect in typhoid toxin packaging was not due to the impaired recruitment of CI-M6PR (Supplementary Figure S5 and Supplementary data set 6". (Note: these data are now shown in Supplementary Figure S16). We assume the reviewer missed these data.

2) COPII can mediate trafficking from the SCV You could do this by checking for co-localization of COPII components with the SCV or the typhoid toxin.

As requested by the Reviewer we have examined the co-localization of Sec23A to the *Salmonella* containing vacuole (SCV). We have found the recruitment of Sec23A to the SCV, particularly at later times after infection. This observation (shown in the revised manuscript as Supplementary Figure S17) is consistent with the involvement of COPII in the packaging and trafficking of typhoid toxin to the extracellular space.

2. You should control for the possibility that secreted toxin undergoes proteolysis in cells lacking M6PR (or other proteins). If this is the case, then important conclusions might be unjustified.

We have assessed this question by examining for the presence of proteases that could potentially degrade typhoid toxin on cell supernatants of *S*. Typhi-infected CI-M6PR cells by exposing purified typhoid toxin to those supernatants. Using this assay, we have found no evidence for the presence of extracellular proteases capable of degrading typhoid toxin (shown in the revised manuscript as Supplementary Figure S4).

3. You should complement the CRISPR KO cell lines described in this study with a plasmid expressing the deleted gene to confirm that the phenotype is not due to a random mutation acquired during the process. Alternatively, you could show same phenotype for independently generated mutants

In each case, we have analyzed at least two independently-generated mutant clones. These data are shown in several additional Supplementary Figures.

4. The interaction between CdtB and Sec23 should be tested as control in the m6pr knockout background.

We have compared the ability of CdtB (a component of typhoid toxin) to form a complex with Sec23 in the presence or absence of CI-M6PR. We found that in CI-M6PR -/- cells, formation of the CdtB/Sec23 complex was abrogated. This observation further supports the premise that the CI-M6PR/COPII complex serves as cargo receptor for typhoid toxin (shown in the revised manuscript as Figure 4j).

Reviewer #1:The typhoid toxin is produced by Salmonella Typhi, the causative agent of typhoid fever in humans. The toxin is secreted into the Salmonella-containing vacuole (SCV) of host cells, from where it must be transported to the extracellular space to carry out its action. In this manuscript by Chang et al. the authors investigate the pathway by which the typhoid toxin is trafficked out of the host cell. They use immunoprecipitation followed by mass spectrometry to identify the cation-independent mannose-6-phosphate (CI-M6PR) as a binding partner for the typhoid toxin and convincingly demonstrate its role as the typhoid toxin sorting receptor. In addition, they show that the Typhimurium effector SseJ blocks recruitment of CI-M6PR to the SCV, confirming that the expression of specific Type III secretion system effectors affects the environment of the SCV. They then use CRISPR/cas9 knockdowns to characterise other host trafficking proteins that are required for toxin export, including COPII, Sar1, Rab11B, Rip11, VAMP7, SNAP23 and Syntaxin 4. These data provide interesting insights into the requirements of host proteins for typhoid toxin export, but a single pathway is not convincingly established, as proteins identified have not been linked together in terms of interaction or dependency with respect to toxin secretion. In addition, the data need to be discussed in the context of established knowledge of eukaryotic trafficking pathways.

We thank the reviewer for the positive comments and the very thorough review of our work. We also understand that the reviewer may feel that our data do not "convincingly establish…a single pathway" responsible for toxin export, and that the reviewer would also like to see our data "interpreted in the context of established knowledge of eukaryotic trafficking pathways". The view of the reviewer is, understandably, "basic cell biology-centric", a view fundamentally shaped by paradigms built through the study of "model systems" (for example EGF, transferrin, or GLU receptor, in the case of endocytic transport). It is important to note, however, that as those of us who have been in the field of "cellular microbiology" from the very early days like to say, "bugs do not read the cell biology textbooks". As the reviewer would certainly understand, this makes the work of studying pathogens much harder because, although knowledge of cell biological process available is extremely useful, often we can't fit our observations into preestablished canons. "Bugs" don't let us do that… There are countless examples of cell biological processes modulated by microbial pathogens in ways that are "unprecedented", and that deviate from established paradigms. This is actually one of the exciting aspects of the study of pathogens because "unprecedented" often times does not mean that bugs do it "their way" but rather, that the study of the canonical systems have missed that aspect of cell biology. In these cases, the studies not only illuminate microbial pathogenesis but also basic cell biology. This study falls somewhere in between… there are mechanistic aspects of typhoid toxin export that fit some previously established paradigm but there are aspects that clearly do not. Consequently, it is natural that the pathway does not neatly fit into a "single pathway". It is "the nature of the beast"… Regardless, when appropriate, we have tried to place our findings in the context of previously described pathways. Again, we thank the reviewer for the thorough review of our work.

1. The results of this work should be discussed in the context of the large body of published work on intracellular trafficking. M6PR intracellular trafficking pathways have been well characterised with information on the signalling sequences found in its cytoplasmic tail, the adaptor proteins they interact with including AP1, AP2, GGAs, PACS^-1^, TIP47 and Rab9 and the specific steps of trafficking pathways that they are involved in. The authors should discuss how their data relates to these pathways. AP1 is known to interact with M6PR and to traffic to and from endosomes and should therefore be included in the screens or at least discussed with a clear reason for its omission. The authors should also comment on the fact that the CI-M6PR normally traffics from the TGN to both endosomes and the plasma membrane, which could provide a plausible pathway.

Ideally, we would have preferred to identify components of the typhoid toxin export pathway through a “genome-wide” screen as we have done for the incoming endocytic pathway using CRISPR/Cas9 [see Chang & Galan, PLoS Pathogens (2019) PMID: 30951565]. Unfortunately, for a variety of reason, it is not possible to implement such screen for the study of typhoid toxin’s export pathway. Consequently, to identify components of typhoid’s toxin export pathway we conducted a more labor-intensive “gene candidate screen", which required the construction of more than 20 different knockout cells using CRISPR/Cas9, amounting to a major undertaking. Using this approach, it is practically impossible to probe every gene reported to be involved in some exocytic pathway. Therefore, we had to prioritize our study focusing on those genes that may be more likely to play a role in typhoid toxin export. For example, the identification of the CI-M6PR sorting receptor allowed us to narrow the set of candidates to investigate. Furthermore, we approached the study in a gradual manner so as to be able to use the information obtained from the examination of some candidates to select additional ones. As alluded by the reviewer, numerous adaptors and coat proteins have been shown to interact with CI-M6PR in different cellular compartments. Among them, and as indicated by the reviewer, are the adaptor proteins AP1 and AP2, which in the context of the sorting of CI-M6PR, have been shown to work in conjunction with clathrin. Since we found no involvement of clathrin in typhoid toxin sorting, we directed our attention to other adaptor/coat components known to be involved in the sorting of the CI-M6PR but in a clathrin-independent manner. We have clarified these issues in the text and expand our discussion of possible link between known CI-M6PR sorting pathways and typhoid toxin secretion.

2. COPII mediates trafficking of proteins between the ER and the Golgi. As such its knockdown should block all export from the ER including that of M6PR. The authors should discuss their data in the context of this information. To be convincing the authors need to rule out any indirect effect caused by blocking ER export and demonstrate clearly that COPII mediates this atypical pathway. For example, it is important to know (1) if COPII is recruited to the SCV membrane in a toxin-dependent manner, (2) if COPII interacts with M6PR and (3) is such an interaction required for toxin export in a way that can be distinguished from general ER export?

As discussed above, we have shown that Sec23A is recruited to the *Salmonella* containing vacuole (SCV), particularly at later times after infection. This observation (shown in the revised manuscript as Supplementary Figure S17) is consistent with the involvement of COPII in the packaging and trafficking of typhoid toxin to the extracellular space. The luminal localization of typhoid toxin, which does not have any transmembrane domain, makes it highly unlikely that it could be involved in the recruitment of CI-M6PR or COPII to the SCV. Therefore, we have not examined this specific question.

3. In order to convincingly establish a pathway, the proteins identified should be linked experimentally. For example, does the interaction of the toxin with Sec23 depend on the presence of M6PR? Can the authors be certain that all identified proteins are involved in direct traffic from the SCV to the plasma membrane? AP4 knockdown also decreased toxin export. Can the authors rule out its involvement in a separate step of the pathway?

We examined whether the interaction between Sec23 and typhoid toxin is dependent on the presence of CI-M6PR (see revised Figure 4j). As discussed above, we found that in the absence of CI-M6PR, the formation of a complex between CdtB (a typhoid toxin component) and Sec23 was abrogated. This observation supports the notion that typhoid toxin, CI-M6PR and Sec23 form a complex. As indicated by the reviewer, the absence of AP4 resulted in a reduced level of typhoid toxin export. However, we did not observe a decrease in the formation of typhoid toxin transport carriers, suggesting a role for AP4 at some step of typhoid toxin export, downstream from its budding from the SCV, or an indirect effect on other aspect of typhoid toxin export. Whether the involvement of AP4 is direct or indirect is not clear at this point. The limitations of our study have now been specifically addressed in the Discussion.

4. The relative toxin export assay is not a convincing measurement. Salmonella infection and perturbation of mannose-6-phosphate receptor traffic have been shown to increase the secretion of proteases (PMID 32514074, 23162002), which could degrade extracellular secreted toxin. Why do the authors use an indirect assay for cell cycle arrest rather than western blot of supernatants? Western blot analysis should be used to determine if lower amounts in extracellular fractions can be rescued by protease inhibition. Can the authors discuss the relevance of an increased relative toxin export as seen after knockdown of AP3B1 and STX-11?

We are aware that it has been reported that infection of cultured cells with *Salmonella* Typhimurium results in a redistribution of the CI-M6PR and the release of proteases to the infection supernatant. However, we stressed that the intracellular biology of *Salmonella* Typhi is substantially different from that of *S*. Typhimurium. For example, although the Rab GTPases Rab29, Rab32, and Rab38 are readily and robustly recruited to the *S.* Typhi-containing vacuole, they are not recruited to the *S*. Typhimurium-containing vacuole. This difference is specifically due to the absence in *S*. Typhi of two T3SS effector proteins (SopD2 and GtgE), which effectively prevent the recruitment of these GTPases to the *S*. Typhimurium-containing vacuole (see for example PMID: 22042847, PMID: 23162001, PMID: 26867180, PMID: 32703879). More relevant to this study, we have shown here that in sharp contrast with *S*. Typhimurium containing vacuole, CI-M6PR is readily and robustly recruited to the *S*. Typhi-containing vacuole, although we did not observe the recruitment of lysosomal hydrolases (i. e. CI-M6PR recruitment was not due to the delivery of S. Typhi to lysosomes) (see Supplementary Figure S1). Therefore, observations made with the study of *S*. Typhimurium vesicle trafficking cannot be directly extrapolated to *S*. Typhi. In any case, we have specifically addressed the reviewer's concern and found no evidence for the presence of extracellular proteases capable of degrading typhoid toxin (shown in the revised manuscript as Supplementary Figure S4).

The amount of toxin exported from infected cells is below the western blot detection limit under our experimental conditions. To be able to detect the toxin by western blot, the experiments would have to be scaled up in a manner that is not practical and susceptible to artifacts. Consequently, we had to use a more sensitive assay to detect typhoid toxin based on its biological activity. We are uncertain about the reasons why AP3B1- and STX-11-defective cells exhibit increased typhoid toxin export. We hypothesize that these proteins may be involved in alternative export pathways that may compete for cellular components involved in typhoid export but we do not have direct proof for this hypothesis.

5. The screen of Typhimurium effectors is interesting – in relation to Typhimurium. But the results do not help understand the features of the Typhi SCV that result in CI-M6PR recruitment. Since the Typhi spiA mutant is defective in packaging toxin into transport carriers (Chang et al. 2016), is this mutant defective in recruiting CI-M6PR to the vacuole? Or is CI-M6PR recruitment simply part of a default phago-lysosomal pathway, as would occur with dead bacteria?

The reviewer is correct and, as expected, the *S*. Typhi *∆spiA* mutant is directed to lysosomes. However, in this compartment, as we have previously shown, the mere presence of CI-M6PR on the SCV is not sufficient for typhoid toxin packaging into vesicle carrier intermediates since packaging requires the intracellular environment of the wild-type SCV (see PMID: 27832592). We understand that there is some confusion on this issue since historically the *S*. Typhi and *S*. Typhimurium vacuoles have been assumed to be similar. However, as we have discussed above, the SCVs in *S*. Typhi and *S*. Typhimurium infected cells are substantially different including differences in their ability to recruit the CI-M6PR, as well as substantial differences in the recruitment of at least 3 Rab GTPases. Stating the obvious, we would like to emphasize that the presence of the CI-M6PR on the wild-type *S*. Typhi SCV is not due to its traffic to lysosomes. Lysosomal hydrolases, for example, are not recruited to the wild-type *S*. Typhi containing vacuole (see Supplementary Figure S1). We have clarified this issue in the text and provided an additional Supplementary Figure to formally document this observation (see Supplementary Figure S1).

Reviewer #2:Their data well justify the authors' conclusions. The article provides an excellent example highlighting the adaptation of an exotoxin co-evolved in the context of the life cycle of an intracellular pathogen producing the toxin. One remaining question not addressed in this article is whether CI-M6PR's role as a sorting receptor for typhoid toxin is shared across different host cells. This question arises because the interaction between CI-M6PR and typhoid toxin PltB is through the glycan, N-acetylneuraminic acid, displayed on the glycosylated sorting receptor(s), suggesting a different glycoprotein might serve as a sorting receptor in other host cells. Nonetheless, this manuscript elegantly determined the detailed sorting and transport process of typhoid toxin occurring in S. Typhi-infected host cells, which is well-justified by the data presented in the manuscript.

We thank the reviewer for the positive assessment of this work.

1. One remaining question not addressed in this article is whether CI-M6PR's role as a sorting receptor for typhoid toxin is shared across different host cells. This question arises because the interaction between CI-M6PR and typhoid toxin PltB is through the glycan, N-acetylneuraminic acid, displayed on the glycosylated sorting receptor(s), suggesting a different glycoprotein might serve as a sorting receptor in other host cells. This can be addressed by investigating other cells relevant to S. Typhi infection. Alternatively, the authors can address this point in the Discussion, perhaps under a separate section "Limitations of this study" to help readers obtain a clearer understanding of the toxin trafficking pathway.

The reviewer is correct, and it remains formally possible that typhoid toxin may engage alternative receptors in different cells. We have presented evidence that CI-M6PR appears to be the main receptor in intestinal Henle-407 and HEK293T cells, two rather different cell lines, which suggest that CI-M6PR is likely the main sorting receptor in several different cells, which is consistent with its ubiquitous expression. We have discussed this and other limitations of our study in the discussion session as suggested by the reviewer.

2. SseJ's role is demonstrated, which is absent in S. Typhi. In some places in the manuscript (i.e., Abstract and Result, page 2, lines 11-13, page 7, lines 16-18), the corresponding descriptions are a bit confusing, sounding like a SPI2 effector produced by S. Typhi plays a key role in the packaging process. Rephrasing some of the descriptions may be considered to help readers get the concept quickly.

We have introduced the suggested editorial changes.

3. Relating to SseJ's role, examining the altered expression and/or localization of OSBP and CI-M6PR on the SCV in various cases tested in the manuscript (i.e., S. Typhi-, S. Typhi+sseJ-, and S. Typhi+sseJS151A-infected host cells) is expected to strengthen the authors' point on the co-evolution of SPI2 effectors and typhoid toxin.

Perhaps the reviewer missed the data, but we did compare the recruitment of CI-M6PR to the SCV of cells infected with wild-type *S*. Typhi and *S*. Typhi expressing either wild-type SseJ or its catalytic mutant SseJ^S151A^. We found that expression of wild type SseJ but not of its catalytic mutant prevented the efficient recruitment of CI-M6PR to the SCV. This led us to conclude that SseJ prevents the recruitment of CI-M6PR in a catalytic-dependent manner (see Figure 3e, Supplementary Figure S9, and Supplementary data set 4).

4. S. Figures S1, S2, S4, S7, and S8: All data from 2-3 independent experiments can be combined and presented as the main figures (as opposed to one representative data for the main figures and all the remaining data for supplementary figures).

We thank the reviewer for the suggestion. However, we prefer the format that we have used to show our data, which more directly demonstrate the reproducibility of the observations.

Reviewer #3:In this manuscript by Chang et al., authors identified the intracellular receptor (cation-independent mannose-6-phosphate receptor, CI-M6PR) of Salmonella Typhi typhoid toxin , which is responsible for sorting this toxin out of the Salmonella-containing vacuole (SCV). This was a curious finding since it was well established by complementary studies using Salmonella Typhimurium that CI-M6PR does not colocalize with the SCV. The authors went on to determine the reason for such differences and identified the S. Typhimurium-specific T3SS effector SseJ as responsible for preventing the co-localization of CI-M6PR in SCVs.The proteins playing a role in xocyticc pathway driving the export of typhoid toxin were also identified. After binding of the typhoid toxin to CI-M6PR in the lumen of the SCV, COPII and SarIB mediate the formation of export carries, which are transported to the plasma membrane in a manner dependent on the GTPase Rab11B and its interacting partner RIP11. Lastly, the fusion of the carriers to the plasma membrane depends on the SNARE proteins VAMP7, SNAP23 and STX4.This work answers a long-standing question in the field, and the author conclusions are well supported by a series of complementary experiments.Some controls are missing in this version of the manuscript, which could provide strength to these findings.In my opinion, the work would greatly beneficiate from live-imaging microscopy analysis showing the trafficking of typhoid toxin within infected or transfected cells. These data could complement the current methodology and make the article more interesting.

We thank the reviewer for the positive assessment of our work. We share the reviewer’s desire to visualize the trafficking of typhoid toxin in live cells. Unfortunately, despite a lot of effort we have not been able to do so, fundamentally, because we can’t get fluorescently tagged versions of typhoid toxin to be efficiently secreted to the bacterial periplasm, despite testing many different tags, some of which reported in the literature to be transported to the bacterial periplasm. This has been an insurmountable challenge that has so far eluded a solution.

1) I think that it is important to complement all the CRISPR KO cell lines described in this study with a plasmid expressing the deleted gene to confirm that the phenotype is not due to a random mutation acquired during the process (either due to the technique itself or the selection of clones within the initial WT population). As all data was performed in HEK293T cells that transfect very well, this could be done relatively easily for CI-M6PR-/-; SEC23B-/-; Sar1B-/-; Rab11B-/-; Rip11-/-; SNAP23-/-; STX4-/-; VAMP7-/-.

In the revised version of the manuscript, we include the analysis of a least two independent clones of the CRISPR/Cas9 generated cell lines with deficiencies in different component of the export pathway (the data are shown in several new Supplementary Figures). We found no significant differences in toxin trafficking between the independent clones.

2) The co-IP experiment shown in Figure 4I could be performed in the CI-M6PR-/- background. If the authors hypothesis is correct, one would expect that the interaction between CdtB and Sec23 (COPII) would be lost.

As suggested by the reviewer, we compared the ability of CdtB (a component of typhoid toxin) to form a complex with Sec23 in the presence or absence of CI-M6PR. We found that in CIM6PR -/- cells, formation of the CdtB/Sec23 complex was impaired (these data are shown in a modified Figure 4). This observation further supports the premise that the CI-M6PR/COPII complex serves as cargo receptor for typhoid toxin.

3) Although discussed in the text, Figure 6C is lacking statistical analysis. This should be included.

The statistical analysis of Figure 6C was present in the figure legend of the original submission (the reviewer must have missed it).

[Editors’ note: what follows is the authors’ response to the second round of review.]

The manuscript has been improved but there are some remaining issues that need to be addressed, as outlined below:They concur that the manuscript is significantly improved and recommended a few points to be addressed (that will not involve new experiments) prior to consideration for acceptance.Reviewer #1:In general, the authors have addressed the main concerns of the reviewers and the manuscript is much approved.As COPII is responsible for ER export there is a high possibility that blocking it through knockout could result in the mislocalisation of many proteins and subsequent off-target effects; this is not fully addressed.In addition, the authors do not discuss the extensive literature on M6PR trafficking including the role of AP1, nor do they conclusively rule out the role of AP1 in the trafficking of M6PR-bound toxin.These points could be mentioned in the ‘limitations of this study’ paragraph in the Discussion.

As suggested by the reviewer we have expanded the discussion of the limitations of our studies to address the reviewer’s concern.